# Denoising as Adaptation: Noise-Space Domain Adaptation for Image Restoration

**Kang Liao, Zongsheng Yue, Zhouxia Wang, Chen Change Loy**
S-Lab, Nanyang Technological University
{kang.liao, zongsheng.yue, zhouxia.wang, ccloy}@ntu.edu.sg
https://kangliao929.github.io/projects/noise-da

## Abstract

Although learning-based image restoration methods have made significant progress, they still struggle with limited generalization to real-world scenarios due to the substantial domain gap caused by training on synthetic data. Existing methods address this issue by improving data synthesis pipelines, estimating degradation kernels, employing deep internal learning, and performing domain adaptation and regularization. Previous domain adaptation methods have sought to bridge the domain gap by learning domain-invariant knowledge in either feature or pixel space. However, these techniques often struggle to extend to low-level vision tasks within a stable and compact framework. In this paper, we show that it is possible to perform domain adaptation via the noise space using diffusion models. In particular, by leveraging the unique property of how auxiliary conditional inputs influence the multi-step denoising process, we derive a meaningful *diffusion loss* that guides the restoration model in progressively aligning both restored synthetic and real-world outputs with a target clean distribution. We refer to this method as *denoising as adaptation*. To prevent shortcuts during joint training, we present crucial strategies such as channel-shuffling layer and residual-swapping contrastive learning in the diffusion model. They implicitly blur the boundaries between conditioned synthetic and real data and prevent the reliance of the model on easily distinguishable features. Experimental results on three classical image restoration tasks, namely denoising, deblurring, and deraining, demonstrate the effectiveness of the proposed method.

## 1 Introduction

Image restoration is a long-standing yet challenging problem in computer vision. It includes a variety of sub-tasks, *e.g.*, denoising (Zhang et al., 2017; Yue et al., 2024), deblurring (Pan et al., 2016; Ren et al., 2020), and deraining (Fu et al., 2017; Wang et al., 2021), each of which has received research attention. Many methods are based on deep learning, typically following a supervised learning pipeline. Since annotated samples are not available in real-world contexts, *i.e.*, degradation is unknown, a common technique is to generate synthetic low-quality data from high-quality images based on assumptions on the degradation process to obtain training pairs. This technique has achieved considerable success but is not perfect, as synthetic data cannot cover all unknown or unpredictable degradation factors, which can vary wildly due to uncontrollable environmental conditions. Consequently, existing methods often struggle to generalize well to real-world scenarios.

Extensive studies have been conducted to address the lack of real-world training data. Some restoration methods improve the data synthesis pipeline to generate more realistic degraded inputs for training (Zhang et al., 2023; Luo et al., 2022). Other blind restoration approaches estimate the degradation kernel from the real degraded input during inference and use it as a conditional input to guide the restoration (Gu et al., 2019; Bell-Kligler et al., 2019). Unsupervised methods (Lehtinen et al., 2018; Shocher et al., 2018; Chen et al., 2023; Ren et al., 2020; Lee et al., 2022) enhance input quality without relying on predefined pairs of clean and degraded images. These methods often use deep internal learning or self-supervised learning, where the model learns to predict clean images directly from the noisy or distorted data itself. In this paper, we investigate the problem assuming the existence of both synthetic data and real-world degraded images. This scenario fits a typical

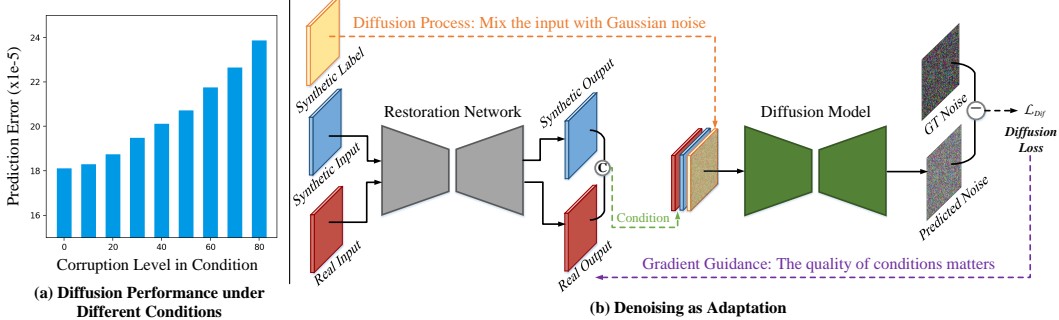

**Figure 1:** (a) The prediction error of a diffusion model is highly dependent on the quality of the conditional inputs. In this experiment, we introduce an additional condition alongside the original noisy input. This condition is the same target image but corrupted with additive white Gaussian noise at a noise level $\sigma \in [0, 80]$. More details can be found in the Appendix A1.1. (b) The restoration network is optimized to provide "good" conditions to minimize the diffusion model's noise prediction error, aiming for a clean target distribution.

domain adaptation setting, where existing methods can be categorized into feature-space (Tzeng et al., 2014; Ganin & Lempitsky, 2015; Long et al., 2015; Tzeng et al., 2015; Bousmalis et al., 2016) and pixel-space (Taigman et al., 2016; Shrivastava et al., 2017; Bousmalis et al., 2017) approaches. Both paradigms have their weaknesses: aligning high-level deep representations in feature space may overlook low-level variations essential for image restoration, while pixel-space approaches often involve computationally intensive adversarial paradigms that can lead to instability during training.

In this work, we present a novel domain adaptation method for image restoration, which allows for a meaningful diffusion loss to mitigate the domain gap between synthetic and real-world degraded images. Our main idea stems from the observation shown in Fig. 1(a). Here, we measure the noise prediction error of a diffusion model conditioned on a noisy version of the target image. The trend in Fig. 1(a) shows that conditions with fewer corruption levels facilitate lower prediction errors of the diffusion model. In other words, "good" conditions give low diffusion loss, and "bad" conditions lead to high diffusion loss. While such a behavior may be expected, it reveals an interesting property of how conditional inputs could influence the prediction error of a diffusion model. Our method leverages this phenomenon by taming a diffusion model conditioned on both the restored synthetic image and restored real image from the restoration network, as shown in Fig. 1(b). Both networks are jointly trained, with the restoration network optimized to provide good conditions to minimize the diffusion model's noise prediction error, aiming for a clean target distribution. Such a goal drives the restoration network to learn to improve the quality of its outputs. After training, the diffusion model is discarded, leaving only the trained restoration network for inference.

While the multi-step denoising process aids the restoration network, a potential shortcut learning could arise: the diffusion model learns to recognize conditions based on their channel index or pixel similarity to noisy synthetic labels, thereby neglecting real data. To mitigate this issue, we propose crucial strategies to fool the diffusion model, making it hard to discriminate between these two conditions. Specifically, we incorporate a channel-shuffling layer into the diffusion model and design a residual-swapping contrastive learning strategy to ensure the model genuinely learns to restore images accurately, rather than relying on easily distinguishable features. These strategies implicitly blur the boundaries between synthetic and real data, ensuring that both contribute effectively during joint training and facilitating their alignment with the target distribution.

To verify the effectiveness of our method, we conducted extensive experiments on three classical image restoration tasks (denoising, deblurring, and deraining), showing promising restoration performance and scalability to different networks. In summary, we make the following contributions:

- Our work represents the first attempt at addressing domain adaptation in the noise space for image restoration. We show the unique benefits from *diffusion loss* in eliminating the gap between the synthetic and real-world data, which cannot be achieved using existing losses.

- To eliminate the shortcut learning in joint training, we design strategies to fool the diffusion model, making it difficult to distinguish between synthetic and real conditions, thereby encouraging both to align consistently with the target clean distribution.

- Our method offers a general and flexible adaptation strategy applicable beyond *specific* restoration tasks. It requires no prior knowledge of noise distribution or degradation models and is compatible with various restoration networks. The diffusion model is discarded after training, incurring no extra computational cost during restoration inference.

## 2 RELATED WORK

**Image Restoration** aims to recover images degraded by factors like noise, blur, or data loss. Driven largely by the capabilities of various networks (Dong et al., 2014; 2015; Zamir et al., 2022; Liang et al., 2021), significant advancements have been made in sub-fields such as image denoising (Zhang et al., 2021; Ren et al., 2021; Guo et al., 2019; Kim et al., 2020; 2024; Fu et al., 2023; Kousha et al., 2022), image deblurring (Kupyn et al., 2018; Suin et al., 2020; Zhang et al., 2019), and image deraining (Jiang et al., 2020; Purohit et al., 2021; Ren et al., 2019; Yang et al., 2017). In image restoration, loss functions are essential for training models. For example, the $L1$ loss minimizes average absolute pixel differences, ensuring pixel-wise accuracy. Perceptual loss uses pre-trained networks to compare high-level features, ensuring perceptual similarity. Adversarial loss involves a discriminator distinguishing between real and synthetic images, pushing the generator to create more realistic outputs. However, the models trained on synthetic images with these conventional losses still cannot escape from a significant drop in performance when applied to real-world domains.

To address the mismatch between training and testing degradations, some supervised image restoration techniques (Zhang et al., 2023; Luo et al., 2022) improve the data synthesis pipeline, focusing on creating a training degradation distribution that balances accuracy and generalization in real-world scenarios. Some methods (Gu et al., 2019; Bell-Kligler et al., 2019) estimate and correct the degradation kernels to improve the restoration quality. Our work is orthogonal to these methods, aiming to bridge the gap between training and testing degradations.

Unsupervised learning methods for image restoration leverage models that do not rely on paired training samples (Huang et al., 2021; Chen et al., 2023; Huo et al., 2023; Chen et al., 2024). Techniques like Noise2Noise (Lehtinen et al., 2018), Noise2Void (Krull et al., 2019), and Deep Image Prior (Ulyanov et al., 2018) exploit the intrinsic properties of images, where the network learns to restore images by understanding the natural image statistics or by self-supervision. These approaches have proven effective in restoration tasks, achieving impressive results comparable to supervised learning methods. However, they often struggle with handling highly complex or corrupted images due to their reliance on learned distributions and intrinsic image properties, which may not fully capture intricate details and show limited generalization to other tasks.

**Domain Adaptation**. The concept of domain adaptation is proposed to eliminate the discrepancy between the source domains and target domains (Saenko et al., 2010; Torralba & Efros, 2011) to facilitate the generalization ability of learning models. Previous methods can be categorized into feature-space and pixel-space approaches. For example, feature-space adaptation methods adjust the extracted features from networks to align across different domains. Among these methods, some classical techniques are developed like minimizing the distance between feature spaces (Tzeng et al., 2014; Long et al., 2015) and introducing domain adversarial objectives (Ganin & Lempitsky, 2015; Tzeng et al., 2017). Aligning high levels of deep representation may overlook crucial low-level variances that are essential for target tasks such as image restoration. In contrast, pixel-space adaptation methods (Liu & Tuzel, 2016; Taigman et al., 2016; Shrivastava et al., 2017; Bousmalis et al., 2017) achieve distribution alignment directly in the raw pixel level, by translating source data to match the "style" of a target domain. While they are easier to understand and verify for effectiveness from domain-shifted visualizations, pixel-space adaptation methods require careful tuning and can be unstable during training. Recent methods (Hoffman et al., 2018; Zheng et al., 2018; Chen et al., 2019) compensate for the limitation of isolated domain adaptation by jointly aligning feature space and pixel space. However, they tend to be computationally demanding due to the need to train multiple networks and the complexity of the cycle consistency loss (Zhu et al., 2017). Different from the above feature-space and pixel-space methods, we propose a new noise-space solution that preserves low-level appearance across different domains within a compact and stable framework.

**Diffusion Model**. Diffusion models (Sohl-Dickstein et al., 2015; Ho et al., 2020; Nichol & Dhariwal, 2021) have gained significant attention in generative modeling. They work by gradually transforming a simple distribution into a complex distribution in a series of steps, reversing the diffusion process.

This approach shows remarkable success in text-to-image generation (Saharia et al., 2022b; Ruiz et al., 2023) and image restoration (Saharia et al., 2022a;c; Yue et al., 2023). Often, conditions are fed to the diffusion model for conditional generation, such as text (Rombach et al., 2022), class label (Ho & Salimans, 2022), visual prompt (Bar et al., 2022), and low-resolution image (Wang et al., 2024), to facilitate the approximation of the target distribution. Some recent works propose to adapt diffusion models for image restoration and its related tasks such as blind JPEG restoration (Welker et al., 2024), open-set image restoration (Gou et al., 2024), and classification of degraded images (Daultani et al., 2024). However, they require the diffusion model in both the training and inference stages. In this work, we show that the diffusion's forward denoising process has the potential to serve as a training proxy task to improve the generalization ability of the image restoration model.

## 3 METHODOLOGY

**Problem Definition.** We start by formulating the problem of noise-space domain adaptation in the context of image restoration. Given a labeled dataset[1] from a synthetic domain and an unlabeled dataset from a real-world domain, we aim to train a model on both the synthetic and real data that can generalize well to the real-world domain. Supposed that $\mathcal{D}^s = \{(\boldsymbol{x}_i^s, \boldsymbol{y}_i^s)\}_{i=1}^{N^s}$ denotes the labeled dataset containing $N^s$ samples from the source synthetic domain and $\mathcal{D}^r = \{\boldsymbol{x}_i^r\}_{i=1}^{N^r}$ denotes the unlabeled dataset with $N^r$ samples from the target real-world domain, where $\boldsymbol{y}^s$ is the clean image, $\boldsymbol{x}^s$ is the corresponding synthetic degraded image, and $\boldsymbol{x}^r$ is the real-world degraded image.

**Image Restoration Baseline.** The restoration network can be formulated as a deep neural network $G(\cdot; \boldsymbol{\theta}_G)$ with learnable parameter $\boldsymbol{\theta}_G$. This network is trained to predict the ground truth $\boldsymbol{y}^s$ from its degraded observation $\boldsymbol{x}^s$ on the synthetic domain. Our domain adaptation is not limited to a specific type of network architecture. One can choose from existing networks such as DnCNN (Zhang et al., 2017), U-Net (Yue et al., 2019), RCAN (Zhang et al., 2018b), and SwinIR (Liang et al., 2021). The approach is also orthogonal to existing loss functions used in image restoration, *e.g.*, $L_1$ or $L_2$ loss, Charbonnier loss (Zamir et al., 2021), perceptual loss (Johnson et al., 2016), and adversarial loss (Wang et al., 2018). To better validate the generality of the proposed approach, we adopt the widely used U-Net architecture $\boldsymbol{f}_\theta(\cdot)$ and the Charbonnier loss $\mathcal{L}_{Res}$, as our baseline. In the joint training, the diffusion model is trained using a diffusion objective, $\mathcal{L}_{Dif}$, while the restoration network is updated using both the $\mathcal{L}_{Res}$ and $\mathcal{L}_{Dif}$. The diffusion model is discarded after training.

### 3.1 NOISE-SPACE DOMAIN ADAPTATION

Ideally, the ground truth images and those restored images by an image restoration model from both synthetic and real-world data should lie in a shared distribution of high-quality clean images. However, attaining such an ideal model that can universally map any degraded images onto the distribution, is exceedingly challenging. Suppose a high-quality image $\boldsymbol{x}$ as a realization derives from a random vector $\boldsymbol{X}$, which belongs to the clean distribution $\boldsymbol{P_X}$. We then define the restored synthetic and real-world outputs from the restoration network as $\hat{\boldsymbol{X}}^s$ and $\hat{\boldsymbol{X}}^r$. In this work, we investigate developing a meaningful diffusion loss to guide the conditional distributions of both synthetic and real-world outputs aligned to the target clean distribution, *i.e.*, $\boldsymbol{P_X} = \boldsymbol{P_{\hat{X}^s}} = \boldsymbol{P_{\hat{X}^r}}$.

Given the commonly adopted case where the ground truth images from the synthetic dataset are available, we first explore adapting the target clean distribution with a perspective of paired data. Without loss of generality, let us consider a synthetic degraded image $\boldsymbol{x}^s$ with its ground truth $\boldsymbol{y}^s$ from the synthetic domain and a real degraded image $\boldsymbol{x}^r$ from the real-world domain. Using the restoration network $G(\cdot; \boldsymbol{\theta}_G)$, we can obtain the restored images $\hat{\boldsymbol{y}}^s$ and $\hat{\boldsymbol{y}}^r$, respectively. Then, based on our observation that the predicted error of a diffusion model is highly dependent on the quality of the conditional inputs, we incorporate a multi-step denoising process as a proxy task into the training process. It employs the predicted images $\hat{\boldsymbol{y}}^s$ and $\hat{\boldsymbol{y}}^r$ as conditions to help the diffusion model fit the clean distribution. Following the notations in DDPM (Ho et al., 2020), we denote the diffusion model as $\boldsymbol{\epsilon}_\theta$ and formulate its optimization to the following objective:

$$\mathcal{L}_{Dif} = \mathbb{E} \left\| \boldsymbol{\epsilon} - \boldsymbol{\epsilon}_\theta \left( \tilde{\boldsymbol{y}}^s | \mathbf{C}(\hat{\boldsymbol{y}}^s, \hat{\boldsymbol{y}}^r), t \right) \right\|_2, \tag{1}$$

where $\tilde{\boldsymbol{y}}^s = \sqrt{\bar{\alpha}_t} \boldsymbol{y}^s + \sqrt{1 - \bar{\alpha}_t} \boldsymbol{\epsilon}$, $\boldsymbol{\epsilon} \sim N(0, \boldsymbol{I})$, $\bar{\alpha}_t$ is the hyper-parameter of the noise schedule, and $\mathbf{C}(\cdot, \cdot)$ denotes the channel-wise concatenation. During the joint training, supervision from the

---

[1]Following the notations in domain adaptation, we use "label" to represent the ground truth image.

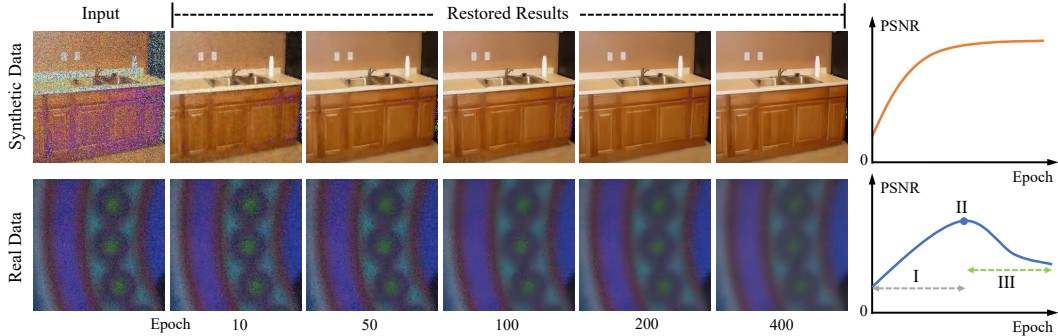

**Figure 2:** During the joint training, the restored synthetic images smoothly converge to the expected distribution over the epochs. However, the model tends to find a shortcut in real data by matching the similarity between the conditions and the paired clean image or remembering the channel index. Consequently, the restoration network learns to corrupt the high-frequency details in real-world images and the diffusion model tends to ignore them.

diffusion loss will back-propagate to the conditions $\hat{y}^s$ and $\hat{y}^r$ if they are under-restored, *i.e.*, far away from the expected distribution. This encourages the restoration network to align $\hat{y}^s$ and $\hat{y}^r$ as closely as possible to the target domain. Particularly, the knowledge "leaked" by the diffusion's input plays an important role, potentially offering degradation-free guidance to help adapt the degraded real-world images into the clean distribution. More discussions are presented in Section A5.2.

The joint training, however, could lead to trivial solutions or shortcuts, as shown in Fig. 2. For example, it is easy to distinguish the synthetic and real-world conditions by the pixel similarity between $\hat{y}^s$ and $\tilde{y}^s$ or the channel index. Consequently, the restoration network will cheat the diffusion network by roughly degrading the high-frequency information in real-world images. As illustrated in Fig. 2 (bottom), we identify three stages in this training process: (I) Diffusion network struggles to recognize which conditions aid denoising as both are heavily degraded, promoting the restoration network to enhance both; (II) Synthetic image is clearly restored and is easy to discriminate from its appearance; (III) The diffusion model tends to distinguish between the conditions, leading it to focus on the synthetic data while ignoring the real-world data.

## 3.2 ELIMINATING SHORTCUT LEARNING IN DIFFUSION

To avoid the above shortcut in the diffusion model, as shown in Fig. 3, we first propose a channel shuffling layer $f_{cs}$ to randomly shuffle the channel index of synthetic and real-world conditions at each iteration before concatenating them, *i.e.*, $\mathbf{C}(f_{cs}(\hat{y}^s, \hat{y}^r))^2$. We show in the experiments that this strategy is important to bridge the gap between synthetic and real data.

In addition to channel shuffling, we devise a residual-swapping contrastive learning strategy to ensure the network learns to restore genuinely instead of overfitting the paired synthetic appearance. Using the ground truth noise $\epsilon$ as the anchor, we construct a positive example $\epsilon^{pos}$ derived from Eq. 1: $\epsilon^{pos} = \epsilon_\theta(\tilde{y}^s | \mathbf{C}(\hat{y}^s, \hat{y}^r), t)$, *i.e.*, the expected noise from the diffusion model conditioned on

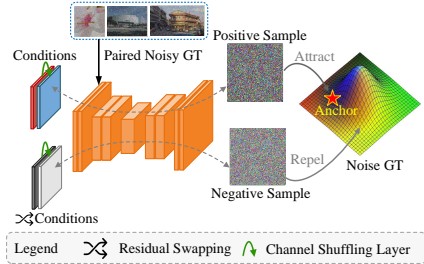

**Figure 3:** The proposed solution to eliminate the shortcut learning in diffusion.

restored synthetic and real-world images. We then swap the residual maps of these two conditions and formulate a negative example $\epsilon^{neg}$ as follows:

$$\epsilon^{neg} = \epsilon_\theta\left(\tilde{y}^s | \mathbf{C}(\hat{y}^{s \leftarrow r}, \hat{y}^{r \leftarrow s}), t\right), \; \hat{y}^{s \leftarrow r} = x^s \oplus \mathcal{R}^r, \; \hat{y}^{r \leftarrow s} = x^r \oplus \mathcal{R}^s, \quad (2)$$

where $\mathcal{R}^s$ and $\mathcal{R}^r$ are the estimated residual maps of the synthetic and real-world images from the restoration network, and $\oplus$ is the pixel-wise addition operator. By swapping the residual of two conditions, we constrain the diffusion model to repel the distance between the wrong restored results

---

$^2$We omit the shuffling operator $f_{cs}$ for notation clarity in the following presentation.

and the expected clean distribution regardless of their context relation. Based on the positive, negative, and anchor examples, a compact residual-swapping contrastive learning can be formulated as:

$$\mathcal{L}_{Con} = \max\left(\|\boldsymbol{\epsilon} - \boldsymbol{\epsilon}^{pos}\|_2 - \|\boldsymbol{\epsilon} - \boldsymbol{\epsilon}^{neg}\|_2 + \delta, 0\right), \tag{3}$$

where $\delta$ denotes a predefined margin to separate the positive and negative samples. In this way, the loss of diffusion model takes the mean of Eq. 1 and Eq. 3, forming the final diffusion loss $\mathcal{L}_{Dif}$. Using the above strategies, we challenge the diffusion model to distinguish between synthetic and real conditions based on trivial solutions, encouraging both to align with the target clean distribution.

In the above formulation, the restored synthetic image of the condition, denoted as $\hat{\boldsymbol{y}}^s$, and the input to the diffusion model, represented as $\tilde{\boldsymbol{y}}^s$, form a pair of data with evident pixel-wise similarity. This similarity can potentially mislead the diffusion model to ignore the real restored image $\hat{\boldsymbol{y}}^r$ in condition as analyzed in Fig. 2. It is important to note that the target distribution encapsulates the domain knowledge of high-quality clean images, including but not limited to the ground truth images in the synthetic dataset. Motivated by this observation, the proposed method can be further extended by replacing the noisy input $\tilde{\boldsymbol{y}}^s$ with $\tilde{\boldsymbol{y}}^c$, defined as $\tilde{\boldsymbol{y}}^c = \sqrt{\bar{\alpha}_t}\boldsymbol{y}^c + \sqrt{1 - \bar{\alpha}_t}\boldsymbol{\epsilon}$, where $\boldsymbol{y}^c$ is randomly sampled from an unpaired extensive high-quality image dataset. This strategy disrupts the pixel-wise similarity between the synthetic condition and the diffusion input, thus enforcing the diffusion model to guide both the synthetic and real conditions predicted by the restoration network at the domain level. We will provide an ablation on this setting in Appendix A4.1.

## 3.3 Training

In the proposed training strategy, the image restoration model is jointly optimized by:

$$\mathcal{L} = \mathcal{L}_{Res} + \lambda_{Dif}\mathcal{L}_{Dif}. \tag{4}$$

Following previous works (Ganin & Lempitsky, 2015), we gradually change $\lambda_{Dif}$ from 0 to $\beta$ to avoid distractions for the main image restoration task during the early stages of the training process:

$$\lambda_{Dif} = \left(\frac{2}{1 + \exp(-\gamma \cdot p)} - 1\right) \cdot \beta, \tag{5}$$

where $\gamma$ and $\beta$ are empirically set to 5 and 0.2 in all experiments, respectively. And $p = \min\left(\frac{n}{N}, 1\right)$, where $n$ denotes the current epoch index and $N$ represents the total number of training epochs.

## 3.4 Discussion

The proposed denoising as adaption is reminiscent of the domain adversarial objective proposed by (Ganin & Lempitsky, 2015). The main difference is that we do not use a domain classifier with a gradient reversal layer but a diffusion network for the loss. We categorize methods like (Ganin & Lempitsky, 2015) as feature-space domain adaptation approaches. Unlike these approaches, we show that denoising as adaptation is more well-suited for image restoration as it can better preserve low-level appearance in the pixel-wise noise space. Compared to pixel-space approaches that usually require multiple generator and discriminator networks, our method adopts a compact framework incorporating only a single additional denoising U-Net, ensuring stable adaptation training. After training, the diffusion network is discarded, requiring only the restoration network for inference. The framework comparison of the above three types of methods is presented in Appendix A2.

## 4 Experiments

**Dataset**. For image denoising, we follow previous works (Zhang et al., 2018a; Zamir et al., 2022) and construct the synthetic training dataset based on DIV2K (Timofte et al., 2017), Flickr2K (Nah et al., 2019), WED (Ma et al., 2016), and BSD (Martin et al., 2001). The noisy images are obtained by adding the additive white Gaussian noise (AWGN) of noise level $\sigma \in [0, 75]$ to the source clean images. We use the training dataset of SIDD (Abdelhamed et al., 2018) as the real-world data. For image deraining, the synthetic and real-world training datasets are respectively obtained from Rain13K (Yang et al., 2017) and SPA (Wang et al., 2019). For image deblurring, GoPro (Nah et al., 2017) and RealBlur-J (Rim et al., 2020) are selected as the synthetic and real-world training datasets, respectively. Please note that we only use the degraded images from these real-world datasets (without

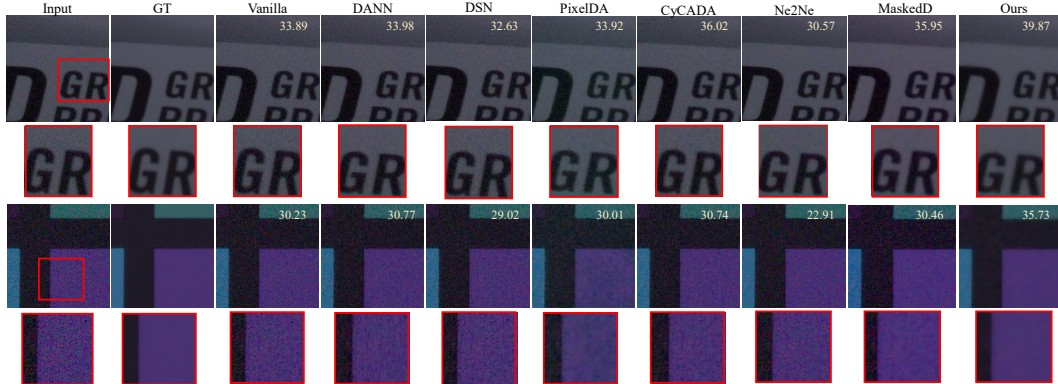

**Figure 4:** Visual comparison of the image denoising task on SIDD test dataset (Abdelhamed et al., 2018). PSNR (dB) is marked for each comparison sample.

**Table 1:** Quantitative evaluation of the image denoising task on SIDD test dataset. *syn*, *real*, *both* denote the model is trained on synthetic, real-world (w/o GT), and both synthetic and real-world (w/o GT) datasets, respectively. $\mathcal{L}_{Res}$, $\mathcal{L}_{Gan}$, $\mathcal{L}_{Ori}$, and $\mathcal{L}_{Dif}$ denotes the pixel-wise restoration loss (*i.e.*, Charbonnier loss), generative adversarial loss, original loss exploited in the paper, and the proposed diffusion loss, respectively.

| Metrics | Vanilla | DANN | DSN | PixelDA | CyCADA | Ne2Ne | MaskedD | Ours |
|---|---|---|---|---|---|---|---|---|
| Space | - | Feature | Feature | Pixel | Feature&Pixel | - | - | Noise |
| Train Data | *syn* | *both* | *both* | *both* | *both* | *real* | *real* | *both* |
| Train Loss | $\mathcal{L}_{Res}$ | $\mathcal{L}_{Res}+\mathcal{L}_{Gan}$ | $\mathcal{L}_{Res}+\mathcal{L}_{Gan}$ | $\mathcal{L}_{Res}+\mathcal{L}_{Gan}$ | $\mathcal{L}_{Res}+\mathcal{L}_{Gan}$ | $\mathcal{L}_{Ori}$ | $\mathcal{L}_{Ori}$ | $\mathcal{L}_{Res}+\mathcal{L}_{Dif}$ |
| PSNR ↑ | 26.58 | 30.09 | 28.40 | 29.24 | 30.81 | 25.61 | 28.51 | 34.71 |
| SSIM ↑ | 0.6132 | 0.7832 | 0.6984 | 0.7611 | 0.8067 | 0.5647 | 0.7196 | 0.9202 |
| LPIPS ↓ | 0.3171 | 0.1348 | 0.2265 | 0.1403 | 0.1256 | 0.3039 | 0.2348 | 0.0903 |

the ground truth) for training purposes. For large-scale unpaired clean images, all images in the MS-COCO dataset (Lin et al., 2014) are used. The test images of the real-world datasets (SIDD, SPA, RealBlur-J) are employed to evaluate the performance of the corresponding image restoration models.

**Training Settings**. To train the diffusion model, we adopt $\alpha$ conditioning and the linear noise schedule ranging from $1e$-6 to $1e$-2 following previous works (Saharia et al., 2022a;c; Chen et al., 2020). Moreover, the EMA strategy with a decaying factor of 0.9999 is also used across our experiments. Both the restoration and diffusion networks are trained on $128 \times 128$ patches, which are processed with random cropping and rotation for data augmentation. Our model is trained with a fixed learning rate $5e$-5 using Adam (Kingma & Ba, 2014) algorithm and the batch size is set to 40.

**Metrics**. The performance of various methods is mainly evaluated using the classical metrics: PSNR, SSIM, and LPIPS. For the image deraining, we calculate PSNR/SSIM using the Y channel in YCbCr color space following existing methods (Jiang et al., 2020; Purohit et al., 2021; Zamir et al., 2022).

## 4.1 COMPARISONS WITH STATE-OF-THE-ART METHODS

We implement the image restoration network using a handy and classical U-Net architecture, which is trained with the proposed noise-space domain adaptation strategy. To validate its effectiveness, we compare the proposed method with previous domain adaptation approaches, including DANN (Ganin & Lempitsky, 2015), DSN (Bousmalis et al., 2016), PixelDA (Bousmalis et al., 2017), and Cy-CADA (Hoffman et al., 2018), covering the feature-space and pixel-space solutions. For the purpose of a fair comparison, we retrained these methods with the same standard settings and datasets. Besides, we also consider some unsupervised restoration methods and representative supervised methods such as Ne2Ne (Huang et al., 2021), MaskedD (Chen et al., 2023), NLCL (Ye et al., 2022), SelfDeblur (Ren et al., 2020), VDIP (Huo et al., 2023), and Restormer (Zamir et al., 2022).

**Comparison Results**. The quantitative and qualitative comparison results are shown in Tab. 1-3 and Fig. 4-5. From the comparison results, the proposed method leads the comparison methods on three image restoration tasks. In particular, previous feature-space domain adaptation methods (Ganin & Lempitsky, 2015; Bousmalis et al., 2016; Hoffman et al., 2018) fail to perceive the crucial low-level

**Table 2:** Quantitative evaluation of the image deraining task on SPA test dataset (Wang et al., 2019).

| Metrics | Vanilla | DANN | DSN | PixelDA | CyCADA | NLCL | Restormer | Ours |
|---------|---------|------|-----|---------|--------|------|-----------|------|
| Space | - | Feature | Feature | Pixel | Feature&Pixel | - | - | Noise |
| Train Data | *syn* | *both* | *both* | *both* | *both* | *real* | *syn* | *both* |
| PSNR ↑ | 33.04 | 32.21 | 33.56 | 30.20 | 32.21 | 20.68 | 34.17 | 34.39 |
| SSIM ↑ | 0.9540 | 0.9443 | 0.9552 | 0.9288 | 0.9442 | 0.8412 | 0.9492 | 0.9571 |
| LPIPS ↓ | 0.0477 | 0.0597 | 0.0512 | 0.0758 | 0.0597 | 0.0967 | 0.0488 | 0.0462 |

**Table 3:** Quantitative evaluation of the image deblurring task on RealBlur-J test dataset (Rim et al., 2020).

| Metrics | Vanilla | DANN | DSN | PixelDA | CyCADA | SelfDeblur | VDIP | Ours |
|---------|---------|------|-----|---------|--------|------------|------|------|
| Space | - | Feature | Feature | Pixel | Feature&Pixel | - | - | Noise |
| Train Data | *syn* | *both* | *both* | *both* | *both* | *real* | *real* | *both* |
| PSNR ↑ | 26.27 | 26.11 | 26.28 | 24.71 | 26.36 | 23.23 | 24.89 | 26.46 |
| SSIM ↑ | 0.8012 | 0.7945 | 0.8003 | 0.7646 | 0.7936 | 0.6699 | 0.7404 | 0.8048 |
| LPIPS ↓ | 0.1389 | 0.1345 | 0.1380 | 0.1583 | 0.1340 | 0.1340 | 0.1589 | 0.1363 |

information and pixel-space domain adaptation methods (Bousmalis et al., 2017; Hoffman et al., 2018) yield inferior results since the precise style transfer between two domains is hard to control during the adversarial training. Moreover, the self-supervised and unsupervised restoration methods (Huang et al., 2021; Chen et al., 2023; Ye et al., 2022; Huo et al., 2023) show noticeable artifacts and limited generalization performance due to some inevitable information loss and hand-crafted designs on specific degradations. By contrast, our method ensures a general and fine domain adaptation in the pixel-wise noise space across various tasks, without introducing unstable training.

**Analysis**. From the above results, we can observe that the proposed method enables noticeable improvements beyond the baseline on the tasks involved with high-frequency noises, such as image denoising. In particular, $+8.13/0.3070$ gains on PSNR/SSIM are achieved. We argue that the target of image denoising naturally fits that of the forward denoising process in the diffusion model. It is more sensitive to other Gaussian-like noises in the pre-sampled noise space. Thus, an intense diffusion loss would be back-propagated if the conditioned images are under-restored, and the restoration network tries to eliminate the noises on both the synthetic and real-world images as much as possible.

## 4.2 ABLATION STUDIES

We conduct ablation studies regarding the sampled noise levels of the diffusion model, determined by the time-step $t$, and the training strategies to avoid shortcut learning, as shown in Tab. 4 and Fig. 6. Concretely, with low noise intensity, *e.g.*, $t \in [1, 100]$, it is easy for the diffusion model to discriminate the similarity of paired synthetic data even when the restored conditions are under-restored. As a result, the shortcut learning occurs earlier during the training process and the real-world degraded image is heavily corrupted by the restoration network, of which most details are filtered. On the other hand, when the intensity of the sampled noise is high, *e.g.*, $t \in [900, 1000]$, the diffusion model is hard to converge and the whole framework has fallen into a local optimum. By sampling the noise from a more diverse range with $t \in [1, 1000]$, the restored results can be gradually adapted to the clean distribution. Moreover, the generalization ability of the restoration network gains further improvement using the designed channel shuffling layer (CS) and residual-swapping contrastive learning strategy (RS), which effectively eliminates the shortcut learning of the diffusion model. Therefore, higher restoration performance on real-world images and more realistic visual appearance can be observed from (d) to (e) and (f) in Tab. 4 and Fig. 6. We also demonstrate that both synthetic data and real data are indispensable for our domain adaptation, excluding each of them would lead to dramatic degradation in real-world performance (shown in the last two rows in Tab. 4). Particularly for excluding the real data, the performance is almost degraded to that of the Vanilla model.

## 4.3 SCALABILITY

**Comparisons.** In this work, we aim to present a general domain adaptation strategy for various restoration tasks, which is scalable to any restoration network. In particular, a basic and lightweight U-Net is used to validate the effectiveness of our method. However, such an architecture essentially

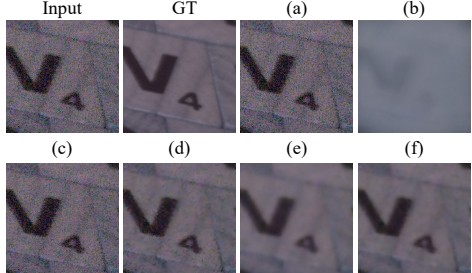

**Figure 5:** Visual comparison of the image deraining and image deblurring tasks on SPA Wang et al. (2019) and RealBlur-J Rim et al. (2020) test datasets. PSNR (dB) is marked for each comparison sample.

| Exp. | Noise Sampling Range [1, 100] | [900, 1000] | [1, 1000] | Strategy CS | RS | PSNR↑ | SSIM↑ |
|---|---|---|---|---|---|---|---|
| (a) | | | | | | 26.58 | 0.6132 |
| (b) | ✓ | | | | | 16.77 | 0.6070 |
| (c) | | ✓ | | | | 27.36 | 0.6590 |
| (d) | | | ✓ | | | 32.07 | 0.8706 |
| (e) | | | ✓ | ✓ | | 32.91 | 0.9082 |
| (f) (Ours) | | | ✓ | ✓ | ✓ | **34.71** | **0.9202** |
| (Only *syn*) | | | ✓ | | | 26.83 | 0.6286 |
| (Only *real*) | | | ✓ | | | 32.60 | 0.8831 |

**Table 4:** Ablation studies of variant networks on the SIDD test image denoising dataset. CS and RS represent the proposed channel shuffling layer and residual-swapping contrastive learning strategies, respectively.

**Figure 6:** Visual comparison results of ablation studies. The complete version (f) of the proposed method achieves the best restoration results with visually pleasant appearances.

limits the upper bound of the restoration performance compared to some recent self-supervised works (Jang et al., 2021; Lee et al., 2022; Jang et al., 2024; Cai et al., 2021) tailored to specific tasks.

Here, we provide experiments to demonstrate higher performance can be achieved using advanced restoration networks with the proposed adaptation strategy. The comparison results are shown in Table 5. In this experiment, we employ a restoration network based on U-Net architecture with deeper layers (named Ours*, the complexity details of different restoration networks are listed in Table A2 of Appendix). The results demonstrate that denoising performance on the SIDD test set has been improved from 34.71 dB to 35.52 dB. Moreover, we show the proposed method can generalize well to other unseen real-world datasets in Fig. 8. These datasets are not encountered during the network's training and fall outside the distribution of the trained datasets.

We believe more powerful restoration networks can enable further improvements, but pursuing extraordinary performance for specific tasks is not the goal of this work.

**Discussion.** Compared to the self-supervised methods (Chen et al., 2023; Ren et al., 2020; Huo et al., 2023; Jang et al., 2021; Lee et al., 2022), our work shows the following unique strengths: it is not bounded to the specific tasks; it is free to the prior knowledge of underlying noise distribution and degradation mode; and it is friendly to the type of preceding restoration networks. We also argue the difference between domain adaptation and self-supervised learning methods: Domain adaptation transfers knowledge from one domain to another with different distributions, improving performance in new, unseen environments. Self-supervised learning, on the other hand, learns from unlabeled data by generating pseudo-labels or exploring the target distribution from the data itself. Both approaches reduce the reliance on large labeled data but address different challenges: domain adaptation focuses on bridging domain gaps, and self-supervised learning leverages data's inherent structure.

**Performance *vs*. Complexity.** We validate the scalability of the proposed method using different variants of U-Net-based restoration networks and other types of architectures, such as the Transformer-based network (Wang et al., 2022). In particular, we classify these networks based on their model sizes and obtain: Unet-T, Unet-S (the model applied in Sec. 4.1), Unet-B, Uformer-T, Uformer-S, and Uformer-B. More details are listed in the Appendix. The quantitative results *vs*. computational costs are shown in Fig. 7. As we can observe, as the complexity increases, the vanilla restoration network (orange elements) tends to overfit the training synthetic dataset and perform worse on the test real-world dataset. In contrast, the proposed method can improve the generalization ability of

| Metrics | C2N | AP-BSN[†] | Ours | Ours* |
|---|---|---|---|---|
| Space | - | - | Noise | Noise |
| Type | SS | SS | DA | DA |
| Train Data | *both* | *real* | *both* | *both* |
| PSNR ↑ | 35.35 | 34.90 | 34.71 | 35.52 |
| SSIM ↑ | 0.9370 | 0.9000 | 0.9202 | 0.9297 |

**Table 5:** Ours* denotes using a more advanced restoration network with deeper layers, trained by our domain adaptation strategy. SS and DA represent the self-supervised and domain adaptation methods, respectively. [†] The asymmetric pixel-shuffle downsampling for the blind-spot network is exploited.

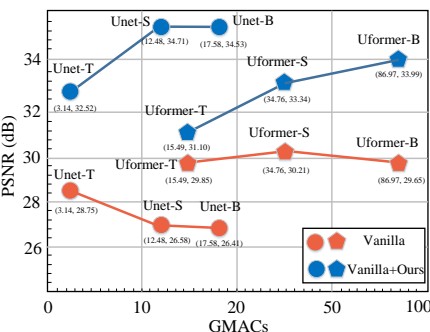

**Figure 7:** Scalability of the proposed method on different network architectures.

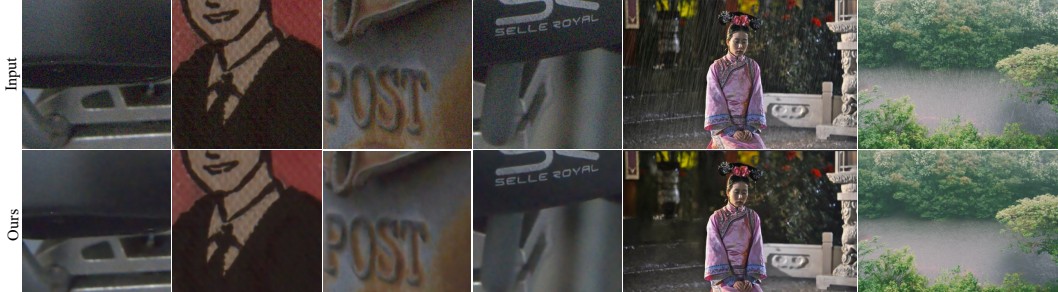

**Figure 8:** Visual results of the proposed method on unseen real-world datasets: the denoising test dataset DND (Plotz & Roth, 2017) and deraining test dataset 'Real-Internet' (Yang et al., 2017).

restoration models with various sizes (blue elements). It is also interesting that for each type of architecture, our method can facilitate better performance as the complexity of the restoration network increases, demonstrating its effectiveness in addressing the overfitting problem of large models.

### 4.4 LIMITATION

The natural mission of the diffusion model is to predict the noises mixed in the input, which are sampled from a high-frequency distribution. Diffusion models excel at capturing and modeling these small-scale variations due to their ability to learn fine-grained details through their denoising process. Thus, more intuitive improvements can be observed in image denoising and deraining tasks, which typically involve high-frequency noises in images. By contrast, artifacts in blurred images, which consist of smooth, gradual changes in intensity, can be less sensitive for diffusion models. They affect larger regions of the image and require the model to correct broad, sweeping distortions rather than fine details. Consequently, diffusion models may struggle to fully restore images with low-frequency noise compared to those with high-frequency noise. We leave it as one of the future work.

### 5 CONCLUSION

In this work, we have presented a novel approach that harnesses the diffusion model as a proxy network to address the domain adaptation issues in image restoration tasks. Different from previous feature-space and pixel-space adaptation approaches, the proposed method adapts the restored results to the target clean distribution in the pixel-wise noise space, resulting in significant low-level appearance improvements within a compact and stable training framework. To mitigate the shortcut issue arising from the joint training of the restoration and diffusion models, we randomly shuffle the channel index of two conditions and propose a residual-swapping contrastive learning strategy to prevent the model from discriminating the conditions based on the paired similarity. Furthermore, the proposed method can be extended by relaxing the input constraint of the diffusion model, introducing diverse unpaired clean images as denoising input. Experimental results demonstrate the effectiveness of our approach over feature-space and pixel-space domain adaptation methods, as well as its scalability surpassing that of self-supervised methods across a range of image restoration tasks.

ACKNOWLEDGMENT

This study is supported under the RIE2020 Industry Alignment Fund – Industry Collaboration Projects (IAF-ICP) Funding Initiative, as well as cash and in-kind contribution from the industry partner(s).

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

APPENDIX

## A1 IMPLEMENTATION DETAILS

### A1.1 CONDITION EVALUATION ON DIFFUSION MODEL

This work is inspired by the beneficial effects that favorable conditions facilitate the denoising process of the diffusion model, as shown in Fig 1(a). In this preliminary experiment, we first condition and train the diffusion model with an additional input in addition to its conventional input. Then, we test the noise prediction performance of this model under different conditions. To be specific, we corrupt the condition by adding the additive white Gaussian noise (AWGN) of noise level $\sigma \in [0, 80]$ to its original clean images, which are performed on 1,000 images in the MS-COCO test dataset (Lin et al., 2014). The noise prediction error of the diffusion model is evaluated using the mean square error (MSE) metric.

### A1.2 COMPARISON SETTINGS

In comparison experiments, we mainly compare the proposed approach with three types of previous methods: domain adaptation methods, including DANN (Ganin & Lempitsky, 2015), DSN (Bousmalis et al., 2016), PixelDA (Bousmalis et al., 2017), and CyCADA (Hoffman et al., 2018); unsupervised image restoration methods, including Ne2Ne (Huang et al., 2021), MaskedD (Chen et al., 2023), NLCL (Ye et al., 2022), SelfDeblur (Ren et al., 2020), and VDIP (Huo et al., 2023); some representative supervised methods which serve as strong baselines in image restoration such as Restormer (Zamir et al., 2022), to comprehensively evaluate generalization performance of different methods. MaskedD (Chen et al., 2023) proposes masked training to enhance the generalization performance of denoising networks, showing the potential to be directly applicable to real-world scenarios. It shares the same goal with our work.

### A1.3 SCALABILITY EVALUATION

To provide a comprehensive evaluation of the proposed method, we apply six variants of the image restoration network in our experiments, including three variants of convolution-based network (Ronneberger et al., 2015): Unet-T (Tiny), Unet-S (Small), and Unet-B (Base); and three variants of Transformer-based network (Wang et al., 2022): Uformer-T (Tiny), Uformer-S (Small), and Uformer-B (Base). These variants differ in the number of feature channels ($C$) and the count of layers at each encoder and decoder stage. The specific configurations, computational cost, and the parameter numbers are detailed below:

- Unet-T: $C$=32, depths of Encoder = {2, 2, 2, 2}, GMACs: 3.14G, Parameter: 2.14M,
- Unet-S: $C$=64, depths of Encoder = {2, 2, 2, 2}, GMACs: 12.48G, Parameter: 8.56M,
- Unet-B: $C$=76, depths of Encoder = {2, 2, 2, 2}, GMACs: 17.58G, Parameter: 12.07M,
- Uformer-T: $C$=16, depths of Encoder = {2, 2, 2, 2}, GMACs: 15.49G, Parameter: 9.50M,
- Uformer-S: $C$=32, depths of Encoder = {2, 2, 2, 2}, GMACs: 34.76G, Parameter: 21.38M,
- Uformer-B: $C$=32, depths of Encoder = {1, 2, 8, 8}, GMACs: 86.97G, Parameter: 53.58M,

and the depths of the Decoder match those of the Encoder.

## A2 DISCUSSION ON DIFFERENT DOMAIN ADAPTATION METHODS

As discussed in Sec. 3.4, we described the effectiveness of the proposed method beyond the previous feature-space and pixel-space domain adaptation methods. We further show their specific framework in Fig. A1. In contrast to previous adaptation methods, our method is free to a domain classifier or discriminator by introducing a meaningful diffusion loss function.

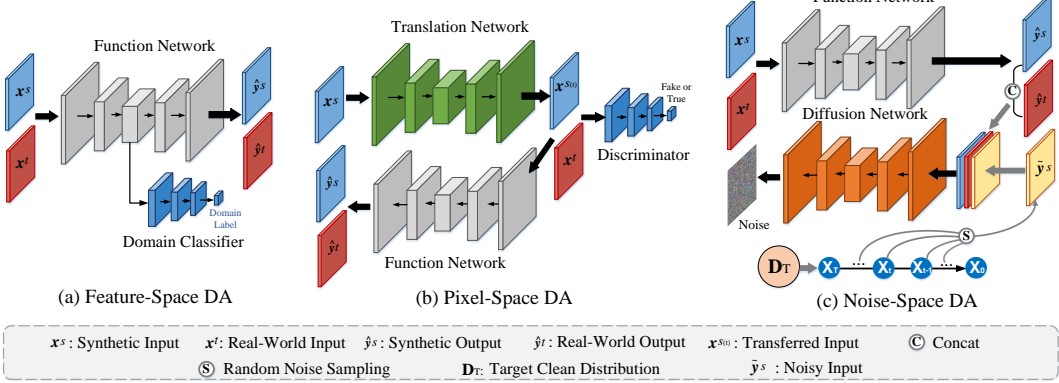

**Figure A1:** Overview of different domain adaptation (DA) approaches. (a) Feature-space DA aligns the intermediate features across source and target domains. (b) Pixel-space DA translates source data to the "style" of the target domain through adversarial learning. (c) The proposed noise-space DA is specifically designed for image restoration. It gradually adapts the results from both source and target domains to the target clean image distribution, via multi-step denoising. Particularly, the function network represents a restoration network in the context of image restoration.

## A3 ADDITIONAL ANALYSIS OF THE ABLATIONS

We provided an ablation study to show the necessity of real data, in which only the synthetic or real data conditions onto the diffusion model. The quantitative results of the SIDD test dataset are listed in Table. 4. It is noteworthy that both synthetic and real data are essential for effective domain adaptation in diffusion models. Omitting either type results in a significant decline in real-world performance. In particular, when real data is excluded, the performance nearly degrades to the level of a Vanilla model. We further analyze the necessity of each condition as follows: (1) Real data typically acts as a "bad" condition that introduces extra noises to the diffusion model, because the restoration network cannot restore it well under the domain gap. Consequently, valid and strong diffusion loss would backpropagate to the restoration network, promoting it learns to provide "good" conditions. As a benefit of the proposed strategies to eliminate the shortcut, the model progressively adapts the real data into the target clean distribution in a multi-step denoising manner. (2) Synthetic data in two conditions can provide useful guidance in the early training stage, ensuring the diffusion model continuously focuses on these condition channels.

## A4 ADDITIONAL COMPARISON RESULTS

### A4.1 EXTENSION

As mentioned in Sec. 3.2, our method can extend to the unpaired condition case by relaxing the diffusion's input with the image from other clean datasets. Thus, the shortcut issue can be potentially eliminated since the trivial solutions such as matching the pixel's similarity between input and condition do not exist. Such an extension keeps the channel shuffling layer but is free to the residual swapping contrastive learning. We show the quantitative evaluation in Tab. A1. The results demonstrate that although the condition and diffusion input are unpaired, our method can still learn to adapt the restored results from the synthetic and real-world domains to the clean image distribution, which also complements the restoration performance of the paired solution in some tasks like deraining and deblurring.

**Table A1:** Quantitative metrics of the proposed method (Ours) and its extension on unpaired condition case (Our-Ex). The results are formed with PSNR/SSIM/LPIPS. The best and second best scores are **highlighted** and underlined.

| Task | Ours | Ours-Ex |
|------|------|---------|
| Denoising | **34.71/0.9202/0.0903** | 34.44/0.8938/0.1064 |
| Deraining | **34.39**/0.9571/0.0462 | 34.20/**0.9587/0.0444** |
| Deblurring | **26.46/0.8048**/0.1363 | 26.44/0.8030/**0.1313** |

**Table A2:** Complexity comparison of the image restoration methods: Parameter (M), GMACs (G).

| Metrics | C2N | AP-BSN | Restormer | Selfdeblur | MaskedD | VDIP | Ours | Ours* |
|---|---|---|---|---|---|---|---|---|
| Parameter | 164.57 | 3.10 | 26.09 | 3.10 | 12.00 | 3.00 | 8.56 | 65.19 |
| GMACs | 280.48 | 60.62 | 35.24 | 23.83 | 188.03 | 23.77 | 12.48 | 22.46 |

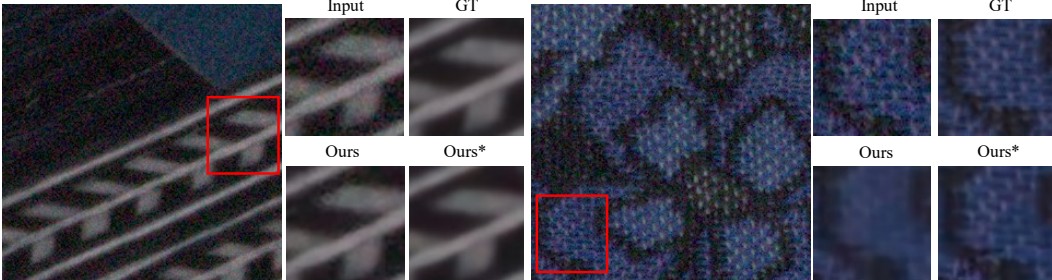

**Figure A2:** Visual results on detailed textures and high-frequency components. The proposed method serves as a general learning strategy for the image restoration task, offering scalability across different restoration networks. It also enables performance improvements as the complexity of the restoration network increases (Ours*).

### A4.2 MORE ADVANCED RESTORATION NETWORKS

As discussed in Sec. 4.3, the proposed domain adaptation method offers strong scalability across various image restoration networks. Additionally, by employing more advanced restoration networks with the proposed denoising as adaptation (Ours*), the performance can be further improved, yielding results that are more perceptually aligned with the ground truth as illustrated in Fig. A2. The complexity comparison of different image restoration networks is listed in Table A2.

### A4.3 ADDITIONAL VISUAL COMPARISON RESULTS

We visualize more comparison results on the image denoising task in Fig. A3, image deraining task in Fig. A4, and image deblurring task in Fig. A5. In particular, we name the proposed method and its extension as 'Ours' and 'Ours-Ex', respectively.

### A4.4 ADDITIONAL VISUAL RESULTS ON OTHER REAL-WORLD DATASETS

To show the generalization ability of the proposed method, we also visualize the restored results of the proposed method on other real-world datasets (Plotz & Roth, 2017; Yang et al., 2017) in Fig. A6, Fig. A7, Fig. A8. Please note that these datasets were not seen during the network's training and fall outside the distribution of the trained datasets.

### A4.5 FAILURE CASE

We show the failure case of our method and comparison methods in Fig A9. Particularly, our method fails to restore the images with challenging degraded distortions such as strong noises and out-of-distribution noises. These real-world degradations induce a significant gap compared with the synthetic dataset, burdening the learning model to effectively adapt the restored results into the clean domain.

### A4.6 ANALYSIS ON TRAINING DYNAMICS AND COMPLEXITY

To demonstrate the impact of the introduced diffusion loss during training, we visualize the related metrics of training dynamics in Fig. A10 (left). It is easy to find that the restoration model trained only with $L1$ loss on the synthetic dataset tends to overfit quickly and performs poorly on the real-world validation set. By contrast, the diffusion loss can effectively guide the restoration model to adapt to the real-world domain in a multi-step denoising manner, consistently improving the restoration performance on the real-world validation set.

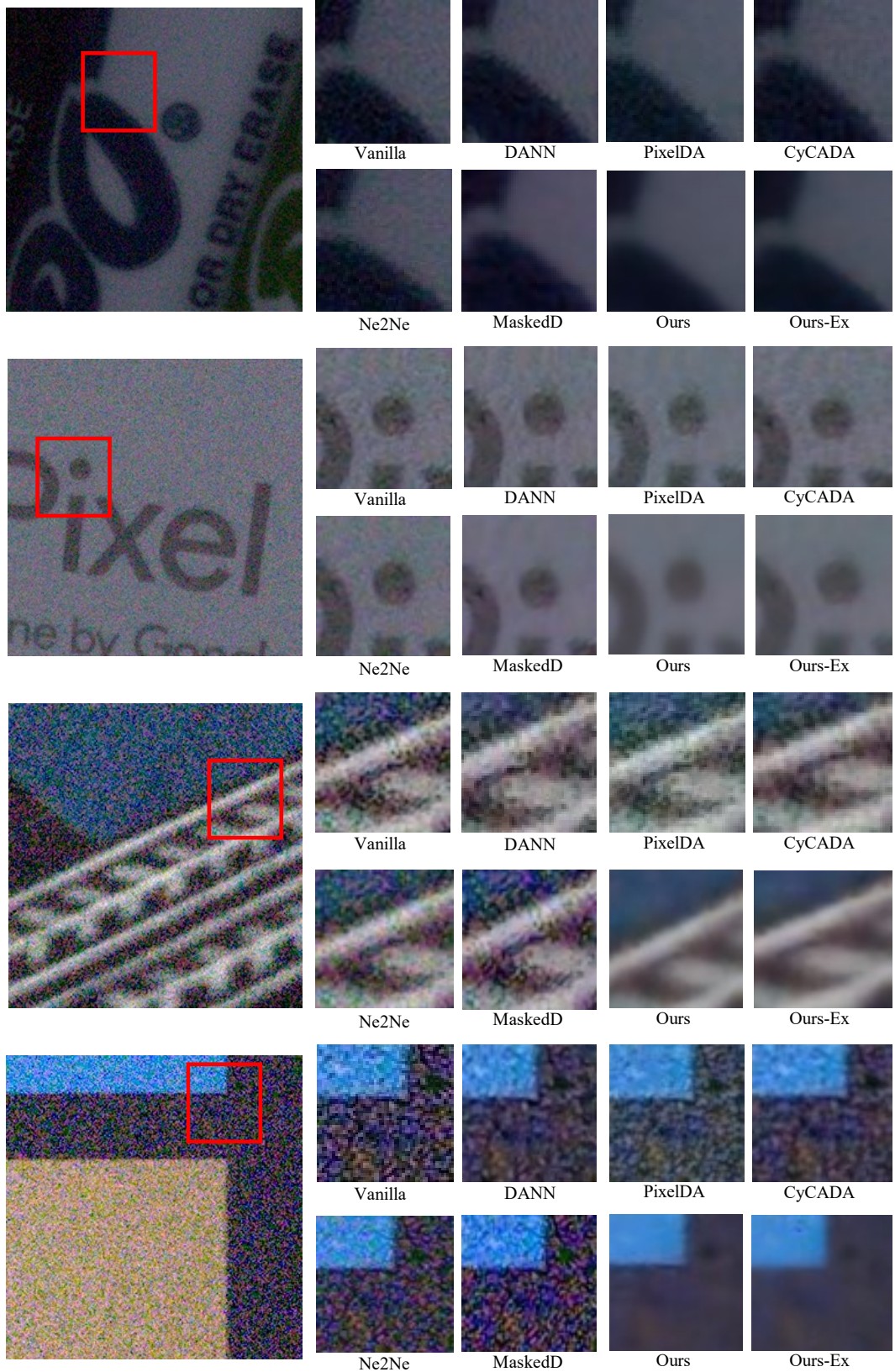

**Figure A3:** Visual comparison of the image denoising task on SIDD test dataset (Abdelhamed et al., 2018).

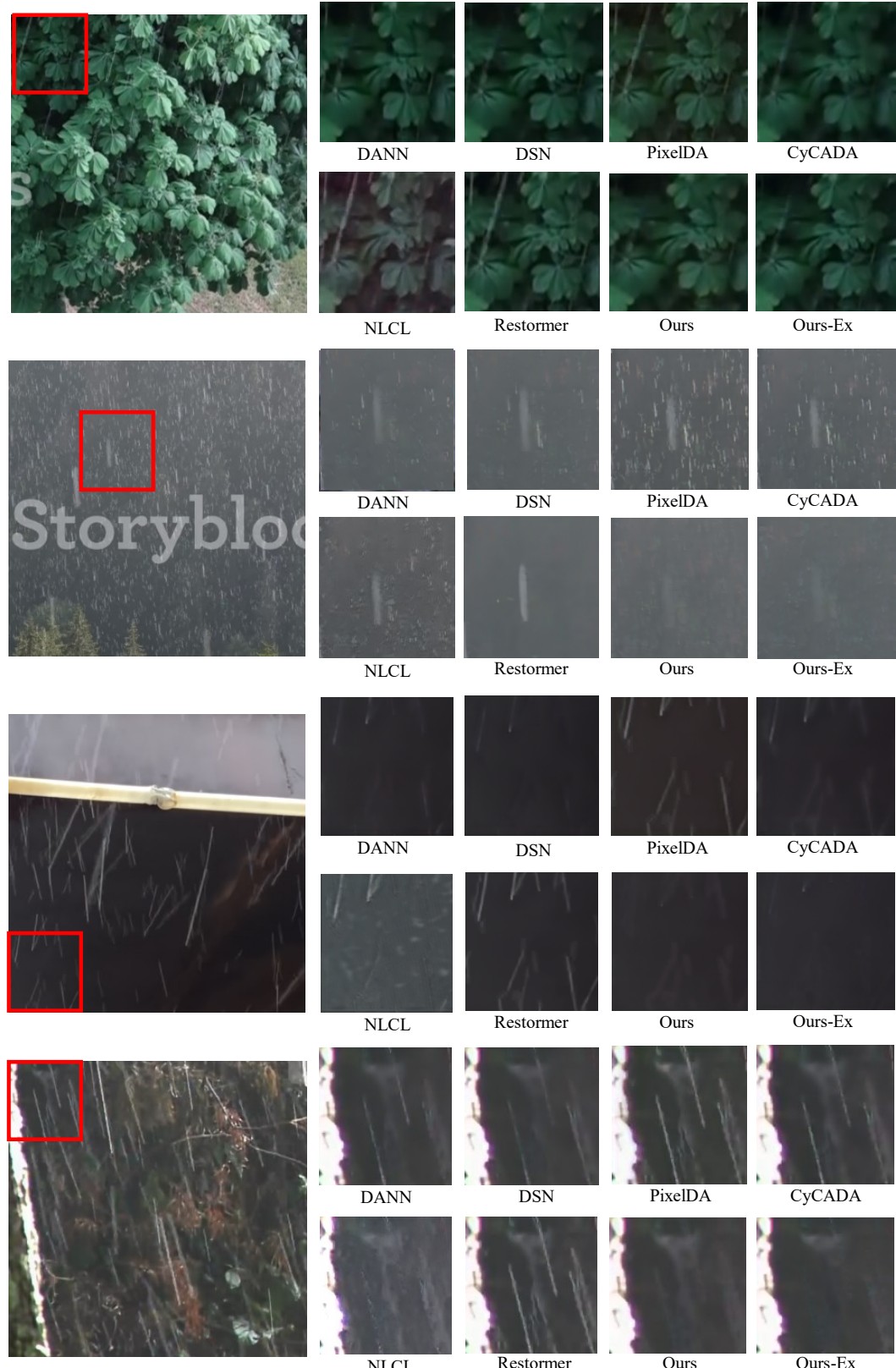

**Figure A4:** Visual comparison of the image deraining task on SPA test dataset (Wang et al., 2019).

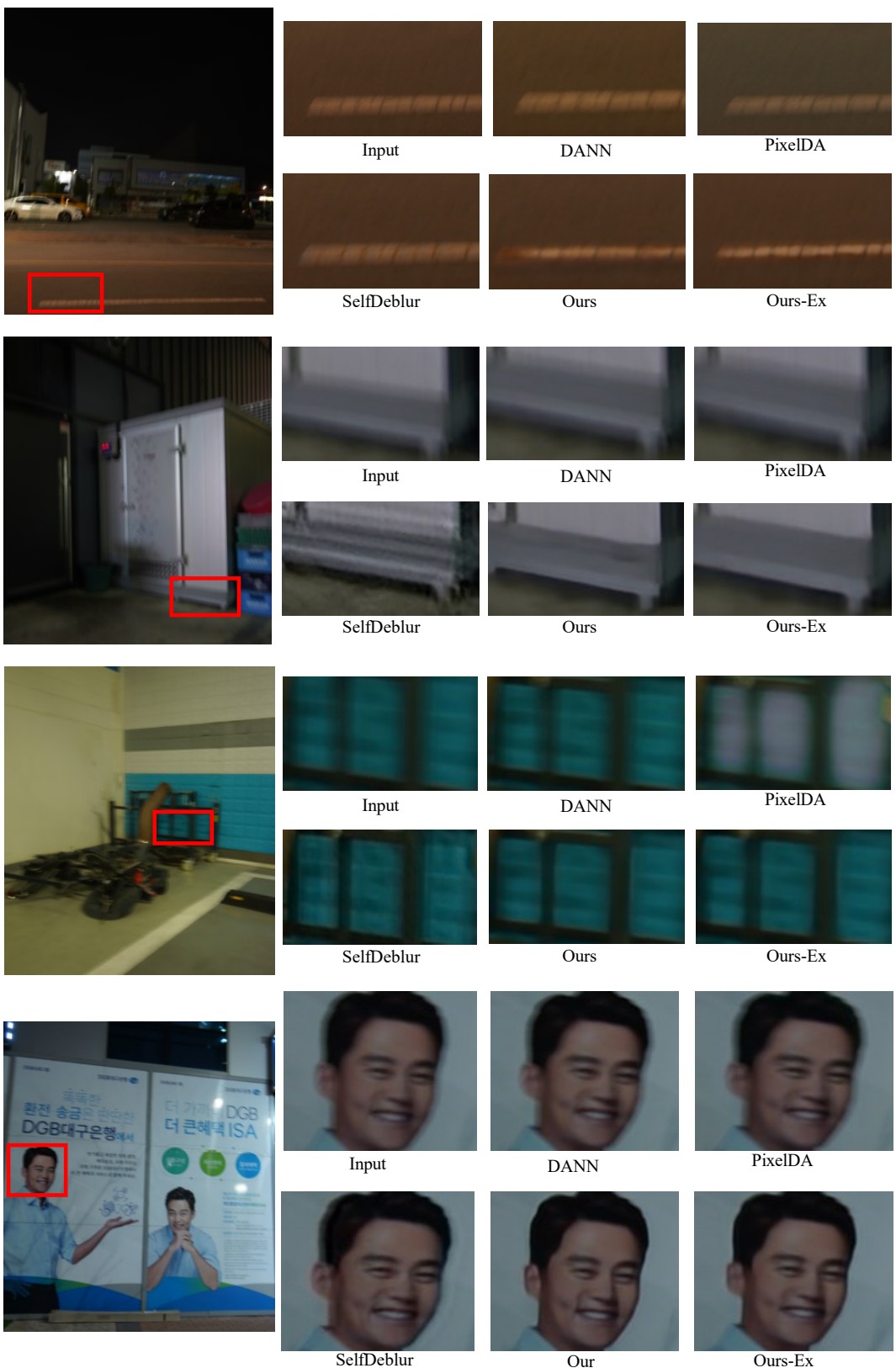

**Figure A5:** Visual comparison of the image deblurring task on RealBlur-J (Rim et al., 2020) test dataset.

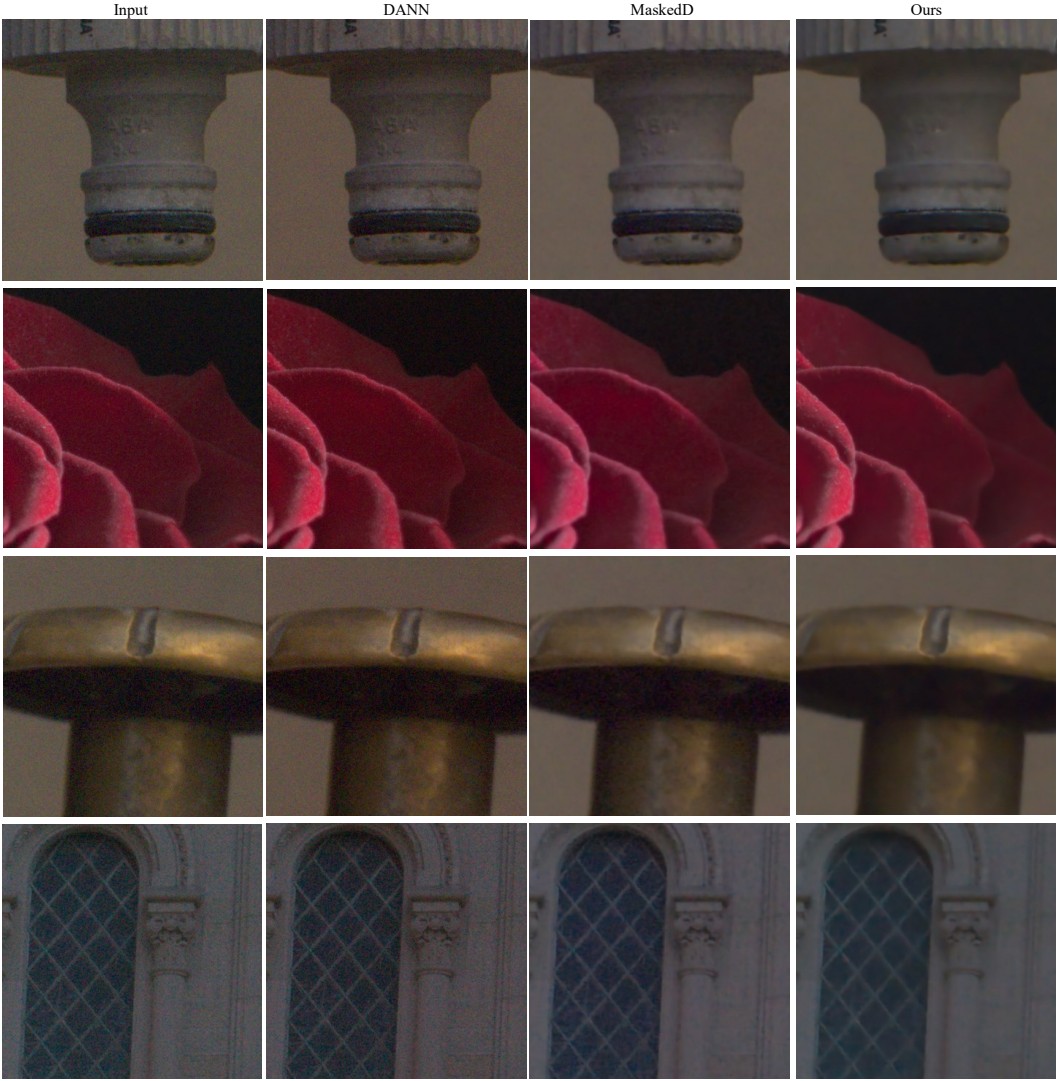

**Figure A6:** Visual results of the proposed method on unseen DND real-world denoising test dataset (Plotz & Roth, 2017).

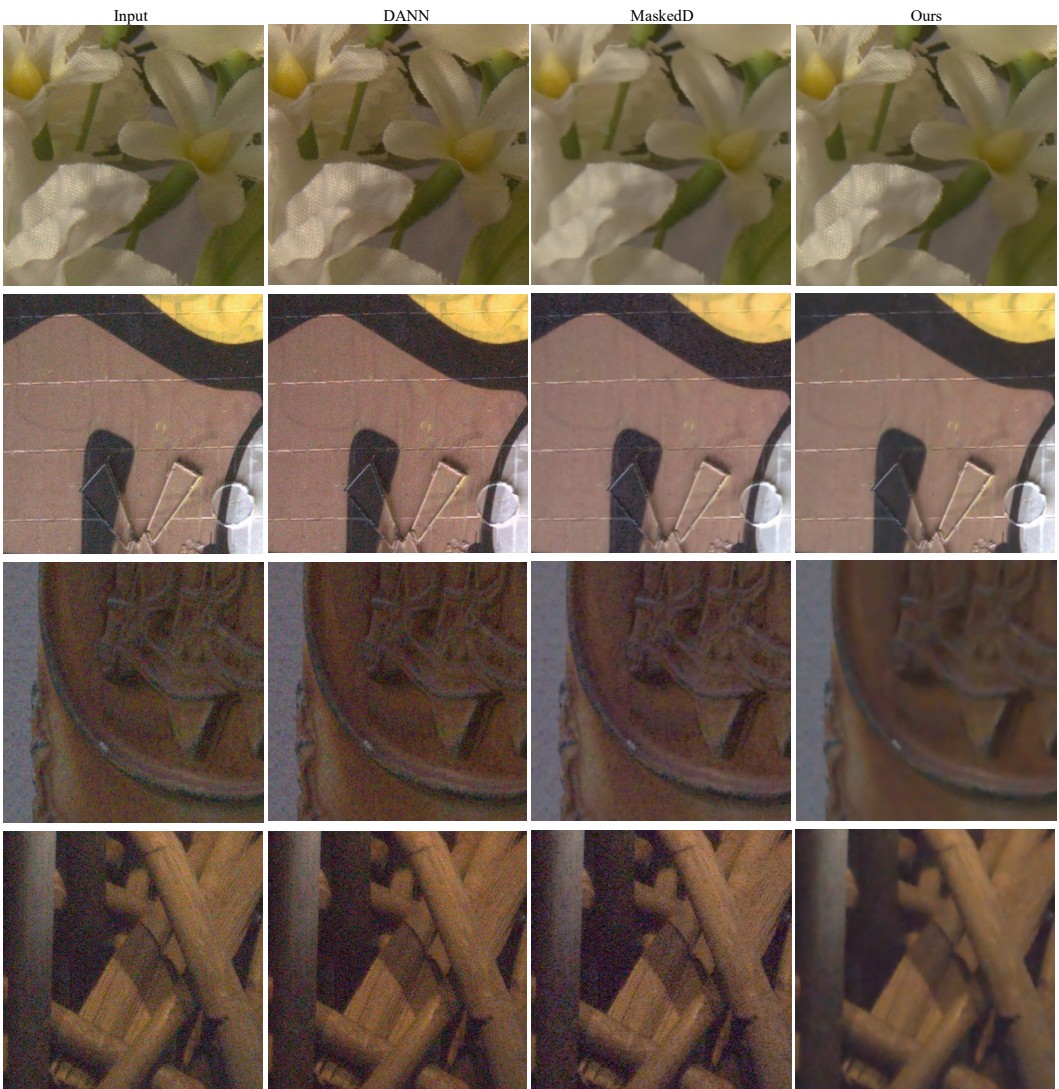

**Figure A7:** Visual results of the proposed method on unseen DND real-world denoising test dataset (Plotz & Roth, 2017).

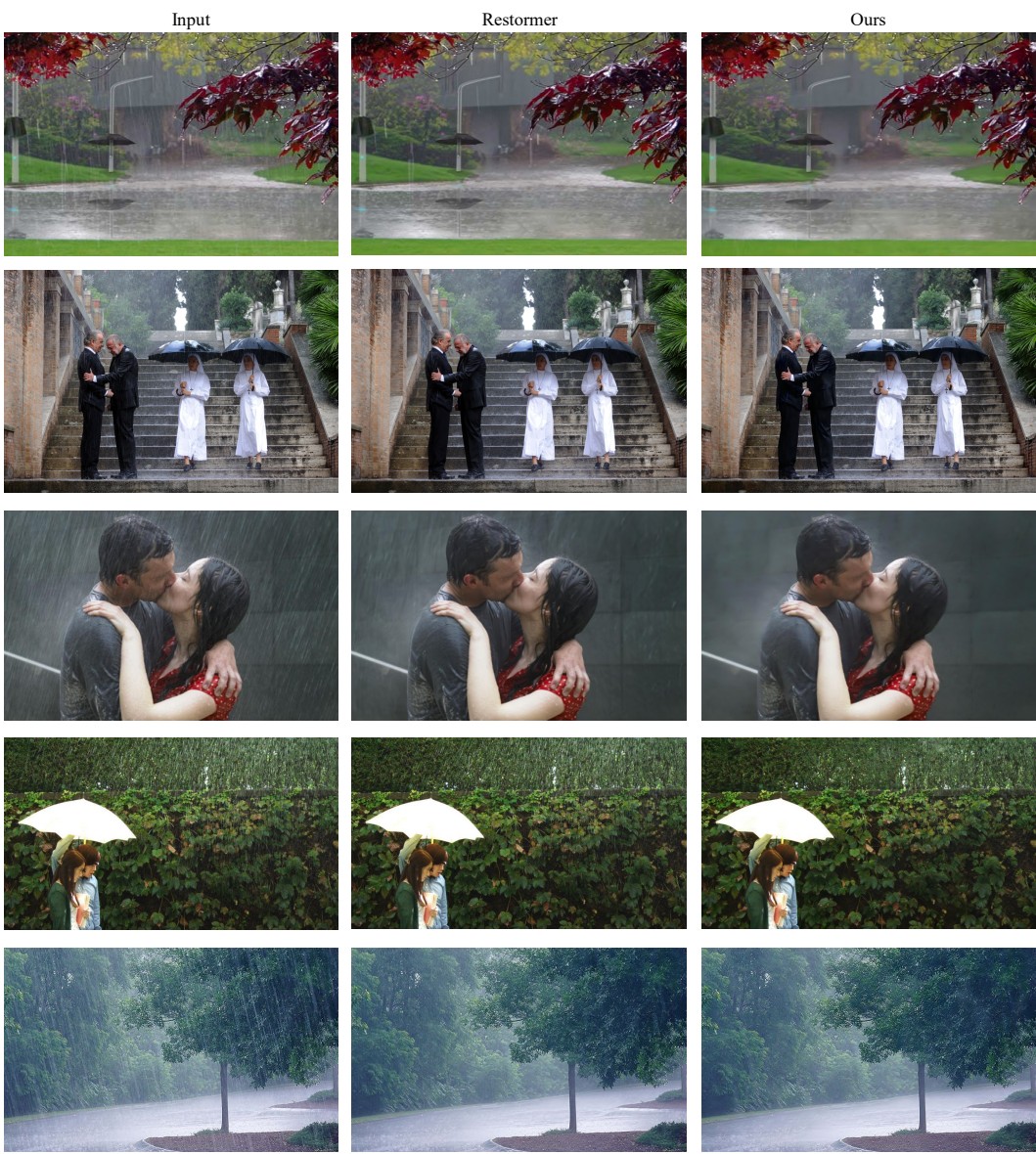

**Figure A8:** Visual results of the proposed method on unseen 'Real-Internet' real-world deraining test dataset (Yang et al., 2017).

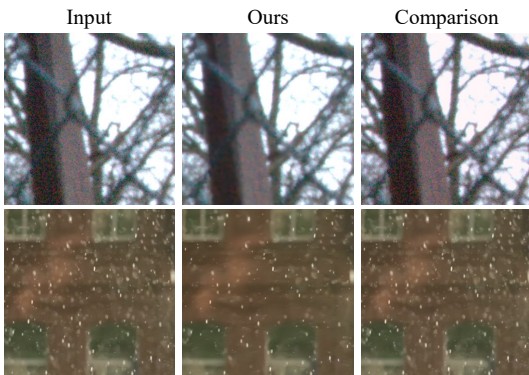

Input    Ours    Comparison

**Figure A9:** Failure case of the proposed method and comparison methods.

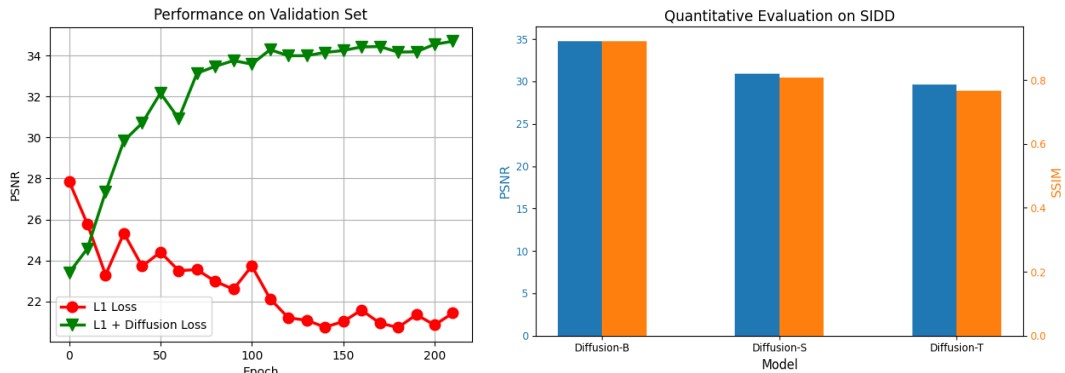

**Figure A10:** Analysis of training dynamics (left) and model complexity (right) of the proposed method.

Moreover, we show the results of validating the diffusion model with different complexities in Fig. A10 (right). To be specific, we classify the diffusion models into three types based on their complexities in each layer: Diffusion-T: [32, 32, 64, 64], Diffusion-S: [32, 64, 128, 128], Diffusion-B: [64, 128, 256, 512] (exploited in this paper). As we can observe, the real-world restoration performance gains further improvements as the complexity of the diffusion model increases, *i.e.*, from Diffusion-T to Diffusion-B. We also provide a deep analysis of diffusion loss in restoration tasks when compared with MAR (Li et al., 2024): MAR models the per-token probability distribution using a small MLP as the diffusion model. It is trained jointly with the AR model to achieve efficient image generation. In particular, the tokens are small in size and represent high-level semantic features. By contrast, our diffusion model serves for the low-level image restoration problem. It directly adapts the restoration results at the dense and wide-scale pixel level, requiring accurate discrimination on the rich texture of images. Therefore, in the context of adapting the restoration model to the real-world domain, the diffusion model cannot be extremely simplified.

|      | **DnCNN** | | **Restormer** | | **SwinIR** | |
|------|---------|--------|---------|--------|---------|--------|
|      | Vanilla | Ours | Vanilla | Ours | Vanilla | Ours |
| PSNR | 25.98 | 30.46 | 24.10 | 33.98 | 30.86 | 34.85 |
| SSIM | 0.5911 | 0.7637 | 0.5194 | 0.9183 | 0.7544 | 0.9153 |

**Table A3:** Performance comparison on SIDD test set of the commonly-used and SOTA restoration models under different training strategies. Vanilla denotes training the model on the paired synthetic datasets and Ours denotes training with the proposed noise-space domain adaptation strategy.

---

**Algorithm 1** Training Process of The Proposed Framework

---

**Input:** Degraded images $\boldsymbol{x}^s$ and ground truth images $\boldsymbol{y}^s$ from synthetic domain $\mathcal{D}^s$, degraded images $\boldsymbol{x}^r$ from real-world domain $\mathcal{D}^r$

**Output:** Restored synthetic image $\hat{\boldsymbol{y}}^s$ and restored real-world image $\hat{\boldsymbol{y}}^r$

1: Initialization on the image restoration model $\boldsymbol{f}_\theta(\cdot)$ and diffusion model $\epsilon_\theta(\cdot)$
2: **repeat**
3:     **for all** $\boldsymbol{x}_i^s$ and $\boldsymbol{y}_i^s \in \mathcal{D}^s$, $\boldsymbol{x}_j^r \in \mathcal{D}^r$ **do**
4:         $\hat{\boldsymbol{y}}_i^s \leftarrow \boldsymbol{f}_\theta(\boldsymbol{x}_i^s), \hat{\boldsymbol{y}}_j^r \leftarrow \boldsymbol{f}_\theta(\boldsymbol{x}_j^r)$
5:         Calculate the restoration loss $\mathcal{L}_{Res}$ with $\hat{\boldsymbol{y}}_i^s$ and $\boldsymbol{y}_i^s$ using Charbonnier loss
6:         Sample the diffusion's input $\tilde{\boldsymbol{y}}_i^s = \sqrt{\bar{\alpha}_t}\boldsymbol{y}_i^s + \sqrt{1 - \bar{\alpha}_t}\epsilon$, $\epsilon \sim N(0, \boldsymbol{I}), t \in [1, T]$
7:         Predict the noise $\tilde{\epsilon} \leftarrow \epsilon_\theta\left(\tilde{\boldsymbol{y}}_i^s | \mathbf{C}(\hat{\boldsymbol{y}}_i^s, \hat{\boldsymbol{y}}_j^r), t\right)$, calculate the diffusion loss $\mathcal{L}_{Dif}$
8:         Update $\epsilon_\theta(\cdot)$ using $\mathcal{L}_{Dif}$
9:         Update $\boldsymbol{f}_\theta(\cdot)$ using Eq.4 with $\mathcal{L}_{Res}$ and $\mathcal{L}_{Dif}$
10:    **end for**
11: **until** Convergence

---

**Algorithm 2** One-Pass Inference of The Proposed Framework

---

**Require:** Degraded image $\boldsymbol{x}^r$, trained image restoration model $\boldsymbol{f}_\theta(\cdot)$
1: $\hat{\boldsymbol{y}}^r \leftarrow \boldsymbol{f}_\theta(\boldsymbol{x}^r)$
2: **return** $\hat{\boldsymbol{y}}^r$

---

### A4.7 VALIDATION ON MORE RESTORATION MODELS

Our work contributes to a novel and general domain adaptation strategy for image restoration, which cannot be replaced by current self-supervised methods. To this end, we further validated our method on the commonly used and SOTA restoration models, such as DnCNN, Restormer, and SwinIR. The quantitative evaluations of these comparison methods are reported in Table. As we can observe, all restoration models trained on synthetic datasets failed to generalize well to the real-world dataset. By incorporating the proposed domain adaptation training strategy, the real-world performance of these models gains significant improvements, demonstrating the favorable generalization and scalability of our method.

## A5 ADDITIONAL TECHNOLOGICAL DETAILS

### A5.1 TRAINING AND INFERENCE

To provide a clear distinction and description of the training and inference stages, we show their detailed processes in Alg. 1 and Alg. 2, respectively. For clarity, we omit the strategies to eliminate the shortcut solutions (exploited in the 7th step of Alg. 1), such as the channel shuffling layer and residual-swapping contrastive learning.

### A5.2 INFORMATION GUIDANCE BETWEEN TWO DOMAINS

Our motivation for this work is derived from an interesting observation: the prediction error of a conditional diffusion model relies on the quality of the conditions (as shown in Fig. 1 (a)). Therefore, guided by the back-propagated diffusion loss, the restoration network is optimized to provide "good" conditions to minimize the diffusion model's noise prediction error, aiming for a clean image distribution. During this joint training, the synthetic GT serves as the denoising target in the diffusion model, which potentially offers realistic textures to help adapt the degraded real-world images into the clean distribution. In other words, the clean knowledge/information "leaked" by the diffusion's input (in a multi-step denoising manner) plays an important role in bridging the gap between different domains.

Generally, the ground truth images and those restored images by a restoration model, whether from synthetic or real-world domains, should reside within a common distribution of high-quality, clean images. However, the appearance of synthetic GT and restored real-world data is unrelated, leading

the diffusion model to exploit a shortcut, overfitting its denoising capability by relying solely on the paired synthetic data. To this end, we further design crucial strategies (*e.g.*, channel-shuffling layer and residual-swapping contrastive learning) to implicitly blur the boundaries between conditioned synthetic and real data and prevent the reliance of the model on easily distinguishable features. As a result, the proactive "leakage" from clean distribution to degraded images can effectively work during the whole training process, consistently improving the restoration performance on real-world images.

