# OpenReview forum: "Denoising as Adaptation: Noise-Space Domain Adaptation for Image Restoration"
_ICLR.cc/2025/Conference — ICLR 2025 Poster_

### Official Review · Reviewer_qhFa · 2024-10-27

**Soundness:** 3
**Presentation:** 3
**Contribution:** 3
**Rating:** 6
**Confidence:** 5

**Summary:**

* This work proposes to address domain adaptation in the noise space for image restoration
* The authors propose channel shuffling and residual-swapping contrastive learning strategy to overcome the shortcut learning issue
* Experiments on image deraining, denoising and deblurring validate the effectiveness

**Strengths:**

* Generalization is a challenging and vital problem in image restoration. The paper proposes to work towards generalization using noise-space adaption.
* The work is well motivated. Especially in Figure 1, the authors find condition of diffusion models can serve as a proxy to discriminate clean distributions. This is interesting for me.
* The paper is well-written. The method is clearly claimed.

**Weaknesses:**

* It seems the paper lacks direct evidence to support diffusion loss is effective. That is, what is the performance if we replace the diffusion loss with perceptual loss, gan loss or nothing. I think this validation is important.
* I suggest to add some visual results of other compared methods in Figure A6-A8.
* Can you provide more details about the used diffusion model? What is the impact of the introduced diffusion loss on training dynamics(eg, loss, accuracy on validation set)? Whether the diffusion model can be extremely simplified (such as a single MLP layer like [1])
[1] Autoregressive Image Generation without Vector Quantization

**Questions:**

Please see weakness

---

> ### Author Response · Authors · 2024-11-22
> **Response to Reviewer qhFa**
>
> > ### W1. Direct evidence to support diffusion loss is effective
>
> We are grateful for the reviewer's valuable suggestions on our validation experiments. Actually, our settings indeed followed such a validation way to evaluate if the proposed diffusion loss was effective. For example, in Table 1-3, the 'Vanilla' denotes the restoration model only trained with L1 loss on synthetic datasets; 'PixelDA' and 'CyCADA' (the mainstream domain adaptation methods) denote the restoration model trained with L1 loss on synthetic datasets and GAN loss on real-world datasets; 'Ours' denotes the restoration model trained with L1 loss and the proposed diffusion loss. To make our validation settings more intuitive, we have highlighted the training loss for each comparison method in Table 1 of the updated paper. Additionally, we have added a new experiment using perceptual loss to replace the diffusion loss. For your convenience,  all quantitative results of the above validations are reported as follows.
>
> |              |    L1 loss    | L1 loss+GAN loss | L1 loss+Perceptual loss | L1 loss+Diffusion loss |
> |:------------:|:-------------:|:------------------:|:-------------------------:|:------------------------:|
> |     PSNR     |    26.58      |       30.81        |          27.27           |          34.71           |
> |     SSIM     |    0.6132     |       0.8067       |          0.6429          |          0.9202          |
>
> From the experimental results, we have two main observations: (1) L1 loss and perceptual loss can effectively minimize average pixel and feature differences for image restoration. However, restoration models trained on synthetic images with these conventional loss functions still cannot escape from a significant drop in performance when applied to real-world domains. (2) GAN loss involves a discriminator distinguishing between real and synthetic images, pushing the generator to create more realistic outputs. However, they tend to be computationally demanding and unstable due to the need to train multiple networks and the complexity of the cycle consistency loss. By contrast, we propose a new noise-space solution that preserves low-level appearance across different domains within a compact and stable framework.
>
> > ### W2. Adding some visual results of other compared methods
>
> Thanks for your kind suggestions. We have added the visual results of other compared methods into Figure A6-A8 in the updated paper. Please refer to the paper for more details.
>
> > ### W3. Can you provide training dynamics and complexity analysis of the diffusion model?
>
> Absolutely. To clearly demonstrate the impact of the introduced diffusion loss during training, we have visualized the related metrics of training dynamics in Fig. A10 (left) of the updated paper. It is easy to find that the restoration model trained only with L1 loss on the synthetic dataset tends to overfit quickly and performs poorly on the real-world validation set. By contrast, the diffusion loss can effectively guide the restoration model to adapt to the real-world domain in a multi-step denoising manner, consistently improving the restoration performance on the real-world validation set.
>
> Moreover, we agree the reviewer’s suggestion on validating the diffusion model with different complexities is an interesting exploration. We have tried this and shown the results in Fig. A10 (right) of the updated paper. To be specific, we classify the diffusion models into three types based on their complexities in each encoder layer: Diffusion-T: [32, 32, 64, 64], Diffusion-S: [32, 64, 128, 128], Diffusion-B: [64, 128, 256, 512] (exploited in this paper).
>
> As we can observe, the real-world restoration performance gains further improvements as the complexity of the diffusion model increases, *i.e.*, from Diffusion-T to Diffusion-B. We also provide a deep analysis of diffusion loss in restoration tasks when compared with MAR [1]: MAR models the per-token probability distribution using a small MLP as the diffusion model. It is trained jointly with the AR model to achieve efficient image generation. In particular, the tokens are small in size and represent high-level semantic features. The trained diffusion model has also been used during the reference sampling stage. By contrast, our diffusion model serves for the low-level image restoration problem. It directly adapts the restoration results at the dense and wide-scale pixel level, requiring accurate discriminations on the rich texture of images. The diffusion model is discarded after training, incurring no extra computational cost during restoration inference. Therefore, in the context of adapting the restoration model to the real-world domain, the diffusion model cannot be extremely simplified. The above discussions have also been added to the updated paper.

---

> > ### Comment · Reviewer_qhFa · 2024-11-25
> > **After rebuttal**
> >
> > Thank you for the detailed rebuttal. The authors address most of my concerns. I raise my score to 6.

---

> > > ### Author Response · Authors · 2024-11-26
> > > **Response to Reviewer qhFa**
> > >
> > > Thank you for raising the rating and for your encouraging comments on our work. We greatly appreciate your recognition of our interesting and well-motivated designs to tame the diffusion model in domain adaptation, as well as the clear clarity of our proposed methods. Your invaluable suggestions will be carefully incorporated into the final version to further enhance the quality of our work.
> > >
> > > Best Regards, Authors

---

### Official Review · Reviewer_vDAP · 2024-10-27

**Soundness:** 3
**Presentation:** 2
**Contribution:** 3
**Rating:** 6
**Confidence:** 4

**Summary:**

- While traditional domain-adaptation methods have focused on adaptation in the feature or pixel-space, this paper emphasizes adaptation in the noise space by leveraging the diffusion model process.
- The training strategy using the diffusion process includes channel-shuffling and residual-swapping contrastive learning to prevent shortcuts learning. Additionally, during inference, only the restoration model is used independently from the diffusion model, making this approach memory-efficient as well.

**Strengths:**

- Domain adaptation in the noise space using the diffusion model process is innovative.
- Although this paper focuses on general restoration tasks such as denoising, deraining, and deblurring, it also potentially explains why the training method using synthetic data struggles to effectively address real data in the denoising problem, and why joint training with both types of data has been ineffective.
- The training strategy to prevent shortcuts is a novel contribution to the restoration task. In fact, shortcut phenomena can occur not only in restoration problems but also in tasks that involve predicting high-frequency components. While it raises the question of whether this strategy can be applied to general CNN networks as well as diffusion models, the paper demonstrates its effectiveness.

**Weaknesses:**

- The performance of the comparison methods used in the experiments is severely degraded. For example, the methods used for comparison in **Table 1** are either outdated or inappropriate. Although relatively recent methods such as **Ne2Ne** and **MaskedD** were used, since these methods do not target denoising using real data, it is questionable whether the results obtained by training these methods on real-world data are reasonable.
- Additionally, since the method presented in the paper uses extra data along with synthetic data for each task, it would be more appropriate to compare it with methods such as **PNGAN**[1*], which use the same training data, as seen in **Table 1**.
- The experiments in **Table 4** demonstrate that the noise sampling range of [1, 1000] is appropriate. Although the model does not require the diffusion process during inference, we believe this range imposes a significant burden during training. Considering this, for example, **DnCNN**[2*] can achieve a gain of approximately 35 dB on the SIDD test dataset with simple training using only L1 loss, even when trained directly on real data.
---
[1*] Cai, Yuanhao, et al. "Learning to generate realistic noisy images via pixel-level noise-aware adversarial training." Advances in Neural Information Processing Systems 34 (2021): 3259-3270.

[2*] Zhang, Kai, et al. "Beyond a gaussian denoiser: Residual learning of deep cnn for image denoising." *IEEE transactions on image processing* 26.7 (2017): 3142-3155.

**Questions:**

- This paper utilizes U-Net and U-Former for simplicity. In my opinion, the key contribution lies in applying the paper’s novel learning strategy to general restoration networks. Therefore, I am curious whether the proposed strategy also performs well on SOTA methods such as **Restormer** or **NAFNet**[1*], and whether these SOTA methods can surpass the performance on real-world data when trained with both real-world and synthetic data.
- Based on the SIDD test set, **NAFlow**[2*], **NeCA**[3*], and **sRGB-Flow**[4*] show much lower performance compared to the restoration achieved with DnCNN using synthetic datasets. According to the claim, the proposed method should be able to bridge the gap between synthetic and real data, thus delivering better performance. It would also be helpful if the paper provided performance results using DnCNN[5*] or other commonly used denoising networks to further validate the effectiveness of the approach.
---
[1*] Chen, Liangyu, et al. "Simple baselines for image restoration." European conference on computer vision. Cham: Springer Nature Switzerland, 2022.

[2*] Kim, Dongjin et al. "sRGB Real Noise Modeling via Noise-Aware Sampling with Normalizing Flows." The Twelfth International Conference on Learning Representations. 2024.

[3*] Fu, Zixuan, Lanqing Guo, and Bihan Wen. "sRGB Real Noise Synthesizing with Neighboring Correlation-Aware Noise Model." Proceedings of the IEEE/CVF Conference on Computer Vision and Pattern Recognition. 2023.

[4*] Kousha, Shayan, et al. "Modeling srgb camera noise with normalizing flows." Proceedings of the IEEE/CVF Conference on Computer Vision and Pattern Recognition. 2022.

[5*] Zhang, Kai, et al. "Beyond a gaussian denoiser: Residual learning of deep cnn for image denoising." IEEE transactions on image processing 26.7 (2017): 3142-3155.

---

> ### Author Response · Authors · 2024-11-22
> **Response to Reviewer vDAP (part 1/2)**
>
> > ### W1. Clarification of the selected comparison methods
>
>  Previous domain adaptation works rarely focused on image restoration. To the best of our knowledge, our work represents the first attempt at addressing domain adaptation in the noise space for various image restoration tasks. Therefore, we mainly considered the classical domain adaptation methods and self-supervised methods as our comparison parts. Particularly, we selected MaskedD in the comparison since this work proposes masked training to enhance the generalization performance of denoising networks. The authors of MaskedD also claimed that "the proposed approach exhibits better generalization ability than other deep learning models and is directly applicable to real-world scenarios". It basically shares the same goal with our work. For more rigorous clarifications, we have highlighted the comparison settings and training data of different works in the "A1.2 Comparison Setting" of the Appendix.
>
> Moreover, as mentioned in your following comments, our work contributes to a novel and general domain adaptation strategy for image restoration, which cannot be replaced by current self-supervised methods. The novelty of taming a diffusion model to achieve this goal has also been acknowledged by other reviewers. Thus, we further validated our method on the commonly used and SOTA restoration models, such as DnCNN, Restormer, and SwinIR, on the SIDD test set. The quantitative evaluations of these comparison methods are reported as follows. Particularly, Vanilla denotes training the model on the paired synthetic datasets, and Ours denotes training with the proposed domain adaptation strategy.
>
> |              | DnCNN          |                | Restormer      |                | SwinIR         |                |
> |--------------|----------------|----------------|----------------|----------------|----------------|----------------|
> |              | Vanilla             | Ours | Vanilla             | Ours | Vanilla             | Ours |
> | PSNR         | 25.98          | 30.46          | 24.10          | 33.98          | 30.86          | 34.85          |
> | SSIM         | 0.5911         | 0.7637         | 0.5194         | 0.9183         | 0.7544         | 0.9153         |
>
>
> As we can observe, all restoration models trained on synthetic datasets failed to generalize well to the real-world dataset. By incorporating the proposed domain adaptation training strategy, the real-world performance of these models gains significant improvements, demonstrating the favorable generalization and scalability of our method. These new comparison experiments have been added to the updated paper.
>
> > ### W2. Considering more comparison methods such as PNGAN
>
> Thanks for suggesting this comparison method. We agree that PNGAN is an excellent self-supervised work tailored for the image denoising task and we have included this work in our paper. However, we must honestly acknowledge that our experimental results do not surpass PNGAN, which can achieve a PSNR of 37.24dB when training the U-Net using its generated noises. This outcome may be anticipated, as our proposed method does not incorporate any prior knowledge of specific restoration tasks, aiming instead to provide a general solution for a wide range of restoration tasks. By contrast, self-supervised methods cannot perform well across different tasks and show limited generalization. Our experiments also demonstrated that by employing a more advanced restoration network, the proposed method can outperform some solid self-supervised learning methods such as C2N and AP-BSN (in Table 5).
>
> We argue the difference between domain adaptation and self-supervised learning: Domain adaptation transfers knowledge from one domain to another with different data distributions, improving performance in new, unseen environments. Self-supervised learning, on the other hand, learns from unlabeled data by generating pseudo-labels or exploring the target distribution from the data itself. Both approaches reduce the reliance on large amounts of labeled data but address different challenges: domain adaptation focuses on bridging domain gaps, and self-supervised learning leverages data's inherent structure.

---

> > ### Author Response · Authors · 2024-11-22
> > **Response to Reviewer vDAP (part 2/2)**
> >
> > > ### W3.  The noise sampling range imposes a significant burden during training. Analysis on compared with DnCNN.
> >
> > We would like to address your concern from the following two aspects.
> >
> > - Compared with previous domain adaptation and self-supervised methods, our method did not introduce significant burdens during training. In Table 4, the experiments verified that a larger sampling range of noise can effectively solve the overfitting problem. For a clear comparison, we show the total GPU hours (on NVIDIA A100 40G GPU) of training different denoising methods as follows. As we can observe, our method is affordable and the whole training can be finished in one day by 4 GPUs. Additionally, as described in Section 3.2 and Appendix A4.1, our method can extend to the unpaired condition case by relaxing the diffusion’s input with the image from other clean datasets. Thus, the shortcut issue can be potentially eliminated since trivial solutions such as matching the pixel’s similarity between input and condition do not exist. Such an extension keeps the channel shuffling layer but is free to the residual swapping contrastive learning. As a result, the training cost can be further reduced as shown in "Ours_ex".
> >
> > |      Training Cost       | Vanilla| DANN | DSN | PixelDA| CyCADA | Ne2Ne | MaskedD | Ours| Ours_ex |
> > |:--------:|:----------------:|:------------------------------:|:--------:|:------------------------------:|:--------:|:--------:|:------------------------------:|:--------:|:--------:|
> > | GPU hours | 35         | 60                      | 130  |  40  |  60  |  80  |  160  | 95  | 66  |
> >
> > - This work targets a challenging domain adaptation problem in image restoration, *rather than overfitting the restoration model on a paired degraded-clean dataset*. It explores how to use the labeled synthetic images to guide restoring the unlabeled real-world images, demonstrating the promising potential for generalization in practical applications. As responded above, most classical restoration models trained on synthetic datasets failed to generalize well to real-world datasets. By incorporating the proposed diffusion loss in noise space, our method is capable of bridging this domain gap effectively, which is friendly to different types of restoration models.
> >
> > > ### Q1. Can the proposed method help SOTA models with generalization performance?
> >
> > Exactly. As mentioned by the reviewer, different from previous self-supervised restoration methods tailored for specific tasks, our contribution is to build a general domain adaptation strategy for various restoration tasks. It requires no prior knowledge of noise distribution or degradation models and is compatible with different restoration networks (as validated in Fig. 7 of the paper, including several variants of U-Net and Uformer). Following your suggestion, we further validated the proposed method with more SOTA restoration models such as Restormer and SwinIR. We also replicated the quantitative evaluation results on the SIDD test set here for your convenience.
> >
> > |              | DnCNN          |                | Restormer      |                | SwinIR         |                |
> > |--------------|----------------|----------------|----------------|----------------|----------------|----------------|
> > |              | Vanilla             | Ours | Vanilla             | Ours | Vanilla             | Ours |
> > | PSNR         | 25.98          | 30.46          | 24.10          | 33.98          | 30.86          | 34.85          |
> > | SSIM         | 0.5911         | 0.7637         | 0.5194         | 0.9183         | 0.7544         | 0.9153         |
> >
> > > ### Q2. A further validate the effectiveness of the approach on DnCNN.
> >
> > Thanks for your suggestion. We have included the above recommended works in our paper. Moreover, we also exploit the commonly used denoising network DnCNN to further validate the effectiveness of our approach. The results demonstrate our method is able to significantly improve the generalization ability of various restoration models. Please refer to the quantitative evaluations in Q1.

---

> ### Comment · Reviewer_vDAP · 2024-11-23
>
> I have a follow-up question as the revised paper has some questions.
> What does $\mathcal{L}1$ in Table 1 refer to? (noise2noise approach? Ne2Ne approach? or what?)
> Also, is $\mathcal{L}_{diff}$ the same as Eq. 4 in your main manuscript?

---

> > ### Author Response · Authors · 2024-11-23
> > **Response to Reviewer vDAP**
> >
> > Thanks for your questions. $\mathcal{L_1}$ denotes the image restoration loss, *i.e.*, $\mathcal{L_{Res}}$. $\mathcal{L_{diff}}$ is the same as the second term of Eq. 4 in the main manuscript. To avoid misunderstanding, we have unified and updated the symbols with a consistent representation in the paper.

---

> > > ### Comment · Reviewer_vDAP · 2024-11-23
> > >
> > > Then, what is the exact formula for the image restoration loss? What I am asking is whether the target image is the ground truth. The caption in the table says “w/o GT,” which makes this confusing.
> > >
> > > If ground truth is not used, the image restoration loss might vary depending on each method (for example, Ne2Ne uses a self-supervised loss in a noise2noise manner). However, this does not seem to be clearly explained.
> > >
> > > What is the exact formula for the L1 loss in your approach in the main text?

---

> > > > ### Author Response · Authors · 2024-11-23
> > > > **Response to Reviewer vDAP**
> > > >
> > > > The exact formula for the image restoration loss is the classical Charbonnier loss, which was presented in Line 188 of the paper. Moreover, "w/o GT" means that the GT of the *real-world dataset* is unavailable during training. Instead, the model is trained using only the paired synthetic dataset (comprising both degraded and GT images) and the degraded real-world images, following the standard configuration of the domain adaptation. These settings have also been marked in the "Train Data" row of Table 1. For the self-supervised methods, the degradation observations of the real-world datasets are used.
> > > >
> > > > For your convenience, we present the exact formula for the $\mathcal{L_{Res}}$ loss as follows.
> > > >
> > > > $$
> > > > \mathcal{L_{Res}} = \sqrt{(I_{{GT}} - I_{{Pre}})^2 + \epsilon^2}
> > > > $$
> > > > where $I_{{GT}}$ is the ground-truth image, and $\epsilon = 10^{-3}$ is a constant in all the experiments.
> > > >
> > > > If the reviewer has any additional questions or concerns, we would be more than happy to address them. Please feel free to leave a comment here. Thank you.

---

> > > > > ### Comment · Reviewer_vDAP · 2024-11-23
> > > > >
> > > > > In the previous rebuttal version, does $L1$ refer to  $L_{\text{res}}$ ?
> > > > > Lastly, let me clarify my main concern, as mentioned earlier: “the image restoration loss might vary depending on each method (for example, Ne2Ne uses a self-supervised loss in a noise2noise manner).”
> > > > >
> > > > > When experimenting with Ne2Ne, did you use  $L_{\text{res}}$ ? If the training data is real, **didn’t you use the loss proposed in the original Ne2Ne method?** In my opinion, the performance gain in methods like Ne2Ne or ZeroShot-N2N is less about the network architecture itself and more about the uniqueness of the loss function. It seems that the loss proposed in the paper should be used.
> > > > >
> > > > > Additionally, based on my experience, with real noise and the SSID test benchmark set, Ne2Ne or ZeroShot-Noise2Noise can typically achieve PSNRs of around 33–34 dB.

---

> > > > > > ### Author Response · Authors · 2024-11-24
> > > > > > **Comments to Reviewer vDAP**
> > > > > >
> > > > > > Thanks for your questions. When experimenting with Ne2Ne, we indeed used the original loss proposed in the paper. It shapes a similar representation with $\mathcal{L_{Res}}$, *i.e.*, the pixel-wise difference between the output and the target. To make the presentation more rigorous, we have updated the training loss of Ne2Ne with "$\mathcal{L_{Ori}}$", which means the original loss used in the proposed method.
> > > > > >
> > > > > > Additionally, in response to your concerns regarding the performance of Ne2Ne, we have thoroughly revisited the experiments and made every effort to retrain the Ne2Ne method on the SIDD dataset. However, the results remain around 26 dB. Regarding the PSNR values of 33-34 dB mentioned in your comments, could you kindly clarify if these were evaluated on the SIDD *raw* data?

---

> > > > > > > ### Comment · Reviewer_vDAP · 2024-11-24
> > > > > > >
> > > > > > > Thank you for your thoughtful and valuable response.
> > > > > > >
> > > > > > > I also revisited the additional experiments regarding the Noise2Noise manner and now agree with the author’s explanation. I apologize for providing incorrect feedback earlier.
> > > > > > >
> > > > > > > Returning to the main content, while I now understand the validity of the author’s experimental approach, I still believe that the descriptions are not sufficiently strict. Although the approach clearly has novelty, the structure of the paper made it challenging to understand.
> > > > > > >
> > > > > > > Specifically, for future paper, I would suggest a clearer distinction and description of the training and inference stages, supported by an appropriate figure. Additionally, as I mentioned earlier, the loss notation is another aspect that requires attention.
> > > > > > >
> > > > > > > Nevertheless, the novelty is undeniable, and since my concerns have been addressed, I will adjust my score to 5 for now. Further adjustments may be made depending on other reviewers’ questions and comments.
> > > > > > >
> > > > > > > Thank you once again for your response.

---

> > > > > > > > ### Author Response · Authors · 2024-11-24
> > > > > > > > **Response to Reviewer vDAP**
> > > > > > > >
> > > > > > > > We appreciate your comments and are grateful for raising your score. In the final version, we will definitely incorporate all your suggestions into the paper, especially in the structure reorganization and training/testing clarification.  We promise the new paper will be updated in the next 1-2 days.
> > > > > > > >
> > > > > > > > In our following discussion with other reviewers, please kindly follow up the updates. And we will help you to summarize their comments and our response here for your reference.
> > > > > > > >
> > > > > > > > Thanks again for your insightful comments.

---

> > > > > > > > ### Author Response · Authors · 2024-11-26
> > > > > > > > **Paper Update Reminder to Reviewer vDAP**
> > > > > > > >
> > > > > > > > Dear Reviewer vDAP,
> > > > > > > >
> > > > > > > > Thank you again for your invaluable suggestions.
> > > > > > > >
> > > > > > > > As promised, we have revised the paper based on your previous feedback. Specifically, we have included detailed workflows for the training and inference stages in Algorithm 1 and Algorithm 2 in the Appendix (page 27). Additionally, we have clarified the loss notations in the caption of Table 1.
> > > > > > > >
> > > > > > > > In the final version, we will ensure that all your suggestions are fully incorporated into the paper. We would greatly appreciate it if you could consider raising your score, should we have adequately addressed your concerns. Thank you once again for your time and effort in reviewing our paper.
> > > > > > > >
> > > > > > > > Best Regards, Authors

---

> > > > > > > > > ### Comment · Reviewer_vDAP · 2024-11-28
> > > > > > > > >
> > > > > > > > > I have read the revised version of your feedback and deeply appreciate your consideration in reflecting on my earlier comments.
> > > > > > > > >
> > > > > > > > > While it would have been ideal to update the figures in the main paper, I understand the constraints related to space limitations and other factors.
> > > > > > > > >
> > > > > > > > > One point that stood out to me during the review process was the shared concern about the PolyU dataset. However, I strongly believe that the issue lies more in the noise characteristics of the PolyU dataset rather than the proposed model itself. It’s possible that similar gains could have been achieved using the SIDD Plus dataset, which would have provided more justification, given the difference in hardware configurations compared to SIDD.
> > > > > > > > >
> > > > > > > > > **In conclusion, I have updated my score from an initial 3 to 6**. While I initially assigned 3 due to concerns with the manuscript’s structure and experiments, after the rebuttal discussion, I found the concept of this paper compelling. If the paper had been better written, I might have considered assigning an 8, which is unfortunate.
> > > > > > > > >
> > > > > > > > > Thank you again for your efforts and for engaging in this discussion.

---

> > > > > > > > > > ### Author Response · Authors · 2024-11-28
> > > > > > > > > > **Response to Reviewer vDAP**
> > > > > > > > > >
> > > > > > > > > > Dear Reviewer vDAP,
> > > > > > > > > >
> > > > > > > > > > We sincerely appreciate you raising your score to 6 and we are truly encouraged by your insightful comments.
> > > > > > > > > >
> > > > > > > > > > We are also grateful that most other reviewers voted their positive scores. Reviewer kUVz claimed most of the concerns have been adequately addressed, but maintained his/her score as 5 due to the “modest gain” tested on a fully unseen dataset PolyU.
> > > > > > > > > >
> > > > > > > > > > In our response, we share a similar perspective to the one reflected in your comments. First, we emphasized that achieving perfect generalization to a fully unseen dataset remains an exceptionally challenging and unsolved problem in the research community. Nonetheless, our method demonstrated an observable performance improvement (+1~2dB) compared to existing approaches.
> > > > > > > > > >
> > > > > > > > > > Moreover, the PolyU dataset’s low noise density and relatively simple noise distribution inherently limit the potential for significant improvement, especially in fully unseen scenarios. To further evaluate our method’s generalization capabilities, we also considered the DND dataset, which features a more complex noise distribution and a broader variety of scenes. This dataset provides a more robust benchmark for assessing generalization. Our experiments on the DND dataset revealed that our method outperformed the comparison methods by a significant margin (+3~4dB), highlighting the promising generalization capability of our restoration model trained using the proposed domain adaptation strategy.
> > > > > > > > > >
> > > > > > > > > > We sincerely believe that the quality of our paper has significantly improved thanks to your professional suggestions. While we regret that the current revisions did not lead to an increase in your score to 8, we deeply appreciate your feedback and will ensure that all your suggestions are thoughtfully incorporated into the final version.
> > > > > > > > > >
> > > > > > > > > > Thank you once again for taking the time and effort to review our paper.
> > > > > > > > > >
> > > > > > > > > > Best Regards, Authors.

---

### Official Review · Reviewer_kUVz · 2024-11-01

**Soundness:** 3
**Presentation:** 2
**Contribution:** 3
**Rating:** 5
**Confidence:** 4

**Summary:**

The introduces an approach to improve the generalization of image restoration models by addressing the domain gap between synthetic and real-world data using noise-space adaptation with diffusion models. The method utilizes diffusion loss, which is optimized during joint training with the restoration network. Key strategies, such as channel-shuffling and residual-swapping contrastive learning, are employed to prevent shortcut learning, ensuring the model effectively restores images rather than relying on trivial cues. The diffusion model is then discarded, leaving a robust restoration network. Experimental results show the method outperforms existing domain adaptation techniques.

**Strengths:**

1. The idea of incorporating diffusion loss in domain adaptation for image restoration is novel.
2. The proposed method can be applied to a variety of restoration tasks, including denoising, deblurring, and deraining, with demonstrated performance gains in the experiments.
3. The use of channel-shuffling and residual-swapping to prevent shortcut learning is well-motivated, and ablation studies show that this technique is crucial for performance improvement.
4. The paper is well-written and easy to understand.

**Weaknesses:**

1. Although ground-truth images from the real dataset are not used for training, degraded images from the same dataset are presented during training, which blurs the distinction between the training and inference stages. To better validate the generalization ability of the proposed framework, driven solely by the inference stage, evaluation on fully unseen datasets (e.g., Nam or PolyU datasets for denoising) should be included.
2. The authors do not provide an analysis of training time, which seems to be a key limitation of the proposed approach. There should be a detailed comparison of training complexity with other models, not just comparisons of inference performance.
3. Ablation studies on the sensitivity to hyperparameters (e.g., beta, gamma) are lacking, particularly as the authors state that these values were chosen empirically.
4. The performance gains compared to Vanilla baseline (trained only with synthetic datasets), while present, are relatively small for deraining and deblurring tasks. Especially for deblurring task, gain is less than 0.2dB.

**Questions:**

Please refer to the Weaknesses for the main questions.

1. How does the model perform on synthetic degradations? Are there significant performance trade-offs due to the domain-adaptation process?
2. Would it be possible to provide quantitative metrics (e.g., PSNR) alongside the visual results (e.g., in Figures 4 and 5) to better illustrate the restoration performance?
3. The paper seems to mainly demonstrate the performance on denoising task, which shows the highest performance gain. Why does the proposed idea fit substantially better for denoising?

---

> ### Author Response · Authors · 2024-11-22
> **Response to Reviewer kUVz (Part 1/2)**
>
> > ### W1. Evaluation on fully unseen datasets
>
> Thanks for helping us enrich the validation experiments. In the previous manuscript, we showed our visual results on fully unseen datasets such as DND and ‘Real-Internet’ in Fig. 8 and Fig. A6-A8. To further validate the generalization ability of the proposed framework, we have followed your suggestion and evaluated the performance on the PolyU dataset. The updated results are shown in the following table. All the evaluated methods are trained on the labeled synthetic dataset and real-world SIDD dataset (only degradation observation).
>
> |            | DANN | CyCADA | Ne2Ne| MaskedD | Ours |
> |:--------:|:----------------:|:------------------------------:|:--------:|:------------------------------:|:--------:|
> | PSNR   | 33.64          | 33.86                       | 32.69  |  33.91  |  34.80  |
> | SSIM   | 0.8001         | 0.8092                     | 0.7609 |  0.8187  |  0.8994  |
>
> > ### W2. An analysis of training time
>
> As suggested, we provide a detailed comparison of training different methods. The evaluated results are reported in the following table. For a clear comparison, we show the total GPU hours (on NVIDIA A100 40G GPU) of training different denoising methods:
>
> |      Training Cost       | Vanilla| DANN | DSN | PixelDA| CyCADA | Ne2Ne | MaskedD | Ours| Ours_ex |
> |:--------:|:----------------:|:------------------------------:|:--------:|:------------------------------:|:--------:|:--------:|:------------------------------:|:--------:|:--------:|
> | GPU hours | 35         | 60                      | 130  |  40  |  60  |  80  |  160  | 95  | 66  |
>
> We also provide a detailed analysis here: Compared with previous domain adaptation and self-supervised methods, our method did not introduce significant burdens during training. As we can observe, our method is affordable and the whole training can be finished in one day using 4 GPUs. Additionally, as described in Section 3.2 and Appendix A4.1, our method can extend to the unpaired condition case by relaxing the diffusion’s input with the image from other clean datasets. Thus, the shortcut issue can be potentially eliminated since trivial solutions such as matching the pixel’s similarity between input and condition do not exist. Such an extension keeps the channel shuffling layer but is free to the residual swapping contrastive learning. As a result, the training cost can be further reduced as shown in "Ours_ex".
>
> > ### W3. Ablation studies on the sensitivity to hyperparameters
>
> We agree the suggested experiments are necessary, and thus we provide the ablation studies on the sensitivity to hyperparameters as follows. In particular, we show the PSNR metric (dB) evaluated on the SIDD denoising test set. The experimental results demonstrate the moderate values for these two hyperparameters achieved the best performance, which means carefully balancing the roles of the restoration model and diffusion mode is crucial during their joint training.
>
> |      beta/gamma      | 1 | 5 | 10|
> |:--------:|:----------------:|:------------------------------:|:--------:|
> | 0.1   | 34.14          | 34.23                      | 34.07  |
> | 0.2   | 34.20         | 34.39                    | 34.11 |
> | 0.5   | 34.09         | 33.94                    | 33.75 |
>
> > ### W4. Analysis of the performance differences across different tasks
>
> The improvement in image deraining (Table 2) shows that the improvement achieved by our method is apparent compared to that of the vanilla model, where +1.35 dB gain can be achieved in the PSNR metric. For the image deblurring task (Table 3), we acknowledge that +0.19 dB improvement is not as noticeable as other tasks; however, the performance still surpasses the classical domain adaptation methods and state-of-the-art self-supervised learning methods.
>
> We also provided a discussion to analyze the different improvements across different restoration tasks as one of our limitations in Section 4.4. The related content has been presented here for your convenience: The natural mission of the diffusion model is to predict the noises mixed in the input, which is usually sampled from a high-frequency distribution. Diffusion models excel at capturing and modeling these small-scale variations due to their ability to learn fine-grained details through their denoising process. Thus, higher improvements can be observed in image denoising and deraining tasks, which typically involve high-frequency noises in images. By contrast, low-frequency noise in blurred images, which consists of smooth, gradual changes in intensity, can be more insensitive for diffusion models. This type of noise affects larger regions of the image and requires the model to correct broad, sweeping distortions rather than fine details. As a result, diffusion models may struggle to fully restore images with low-frequency noise compared to those with high-frequency noise. We leave it as one of the future works.

---

> > ### Author Response · Authors · 2024-11-22
> > **Response to Reviewer kUVz (Part 2/2)**
> >
> > > ### Q1. How the model performs on synthetic degradations
> >
> > We provided the evaluation results as follows. Specifically, we report the differences between PSNR values obtained by our method compared with the vanilla model only trained on synthetic data. As we can see, our method performs well on synthetic datasets without significant performance degradations while showing promising improvements on the real-world dataset. We provide further analysis here: Our domain adaptation strategy simultaneously trains the image restoration network using L1 loss (minimizing the restoration difference on synthetic paired results) and diffusion loss (minimizing the domain gap between synthetic data and real data). Therefore, our method can effectively perform real-world adaptation while avoiding the performance trade-offs between these two domains.
> >
> > |                | Urban100       |               |               | CBSD68         |               |          | SIDD  |
> > |----------------|----------------|---------------|---------------|----------------|---------------|-------|-------|
> > |                | sigma_15       | sigma_25      | sigma_50      | sigma_15       | sigma_25    | sigma_50   | Real  |
> > | PSNR Gains (dB)| -0.36          | -0.32         | -0.26         | -0.20          | -0.12         | -0.13         | +8.13 |
> >
> > > ### Q2. Quantitative metrics alongside the visual results
> >
> > Thanks for this kind suggestion. We have updated the quantitative metrics alongside the visual results in Fig.4 and Fig. 5. Please refer to the updated paper for more details.
> >
> > > ### Q3. Why does the proposed idea fit substantially better for denoising?
> >
> > As discussed in W4, we have provided analysis for different performances across different tasks. We would like to further summarize our analysis from two aspects: (1) First, the diffusion model is an essential denoiser. Its natural mission is to predict the noises mixed in the input, which is usually sampled from a high-frequency distribution. Diffusion models excel at capturing and modeling these small-scale variations due to their ability to learn fine-grained details through their denoising process. Thus, the involvement of other types of high-frequency noise in conditions would mislead the diffusion model and yield a strong loss. (2) Second, the L2 loss used in training diffusion models is often more sensitive to high-frequency components. It encourages the model to focus on accurately reconstructing fine details, thus performing well in scenarios with high-frequency noise.

---

> ### Comment · Reviewer_kUVz · 2024-11-24
>
> Thank you for your detailed and thoughtful response. While most of my initial concerns have been adequately addressed, the issue regarding the vague distinction between the training and inference stages remains only partially resolved.
>
> In particular, the observed performance discrepancy on unseen datasets raises concerns. Unlike SIDD, which demonstrates significant improvement (+4–6 dB), the framework achieves only a modest gain on the on fully unseen Poly dataset (+1–2 dB). This observation strongly suggests that the adaptation process relies heavily on the noise distributions encountered during training, thereby limiting the practical applicability and generalizability of the proposed framework.
>
> As a result, I have decided to retain my original score, as the effectiveness in broader, unseen scenarios appears constrained.
> I hope my feedback provides valuable guidance for improving this aspect in future work.
>
> Best regards,
> Reviewer KUVz

---

> > ### Author Response · Authors · 2024-11-24
> > **Response to Reviewer kUVz**
> >
> > Thank you for your valuable feedback. We would like to highlight that generalizing to a fully unseen dataset presents a significantly greater challenge compared to the classical domain adaptation problem. The PolyU dataset differs substantially from the SIDD dataset in terms of noise density and scene diversity. Consequently, we kindly disagree with the characterization of a +1-2 dB performance improvement on the fully unseen dataset as a *modest gain*. In fact, this improvement surpasses the performance of previous methods.
> >
> > If possible, could you kindly provide references to methods that achieve a *significant gain* on such a fully unseen dataset? We would like to compare our approach with them in the following experiments.
> >
> > Best regards, Authors.

---

> > > ### Comment · Reviewer_kUVz · 2024-11-25
> > >
> > > I agree that achieving a +1–2 dB performance improvement is indeed challenging. However, my primary concern lies in the relatively low performance gain observed on fully unseen datasets, particularly when compared to the results on SIDD, where the framework had access to noisy images during the training stage. This discrepancy suggests that the framework's **domain adaptation** capability is not adequately demonstrated by the experiments presented in the main paper, as they primarily focus on highlighting performance gains within the SIDD dataset. Further investigation into the framework's generalization ability across diverse noise distributions would strengthen the paper's claims.
> > >
> > > Best regards, Reviewer KUVz

---

> > > > ### Author Response · Authors · 2024-11-27
> > > > **New Experiments on More Fully Unseen Datasets**
> > > >
> > > > Dear Reviewer kUVz,
> > > >
> > > > Thank you again for your valuable suggestion regarding the generalization performance on fully unseen datasets. As mentioned in our previous responses and experiments, while our method outperforms comparison methods on the PolyU dataset, the improvement space is inherently limited due to its relatively low noise density.
> > > >
> > > > To further evaluate our generalization performance, we have considered other fully unseen datasets, such as the DND dataset. The DND dataset presents a more complex noise distribution and encompasses a wider variety of scenes, making it a suitable benchmark for assessing generalization. The quantitative results from this evaluation are listed as follows.
> > > >
> > > > |            | DANN | CyCADA | Ne2Ne| MaskedD | Ours |
> > > > |:--------:|:----------------:|:------------------------------:|:--------:|:------------------------------:|:--------:|
> > > > | PSNR   | 27.74          | 26.33                      | 29.37  |  33.70  | 37.10  |
> > > > | SSIM   | 0.6913         | 0.6447                     | 0.6989 |  0.8605 |  0.9298  |
> > > >
> > > > From the experimental results, we observe that our method significantly outperforms the comparison methods by a large margin. This highlights the promising generalization capability of the restoration model trained with the proposed domain adaptation strategy.
> > > >
> > > > If you have any additional questions or concerns, we would be delighted to address them. Please feel free to leave a comment at any time. Thank you once again for your thoughtful review and constructive feedback.
> > > >
> > > > Best Regards, Authors.

---

> > > > ### Author Response · Authors · 2024-12-01
> > > > **Please kindly let us know if your question has been addressed**
> > > >
> > > > Dear Reviewer kUVz,
> > > >
> > > > Thank you for your valuable feedback and insightful comments. We are glad that most of your concerns have been adequately addressed. As the discussion period ends soon, we kindly ask if our responses have effectively addressed your questions on the generalization of this work.
> > > >
> > > > To assist your review process, we have taken the initiative to compile a summary of our responses along with the related comments from other reviewers: In the previous rebuttal, we explained that the PolyU dataset’s low noise density and relatively simple noise distribution inherently limit the potential for significant improvement, especially in fully unseen scenarios. Such a conclusion was also supported by Reviewer vDAP, and Reviewer vDAP strongly believed the issue lies more in the noise characteristics of the PolyU dataset rather than the proposed model itself.
> > > >
> > > > Furthermore, we have added a new generalization experiment on the DND dataset. This dataset provides a more robust benchmark for assessing generalization due to its complex noise distributions and a broader variety of scenes. Experiments on the DND dataset revealed that our method outperformed the comparison methods by a significant margin (+3~4dB), highlighting the promising generalization capability of the restoration model trained using the proposed domain adaptation strategy.
> > > >
> > > > Should you have any further inquiries or require additional clarification, we would be more than happy to assist you. We greatly appreciate your insightful review and valuable feedback.
> > > >
> > > > Best Regards, Authors.

---

> ### Author Response · Authors · 2024-11-26
> **Response to Reviewer kUVz**
>
> Dear Reviewer kUVz,
>
> We sincerely thank you once again for your insightful feedback. Perfectly generalizing a network's restoration capacity to fully unseen domains is indeed a meaningful and practical direction. However, given the significant domain gap, it remains an unsolved and open problem in the low-level vision community. To the best of our knowledge, no existing methods have achieved this ideal target.
>
> Our work represents the first attempt to address domain adaptation specifically in the noise space for image restoration. We are also grateful that this contribution and its novelty have been recognized by you and other reviewers. Moreover, the performance comparison on the suggested fully unseen dataset highlights our noticeable performance gain over other methods.
>
> Nevertheless, we honestly acknowledge that the performance improvement on the SIDD dataset is relatively higher than that on the PolyU dataset. This discrepancy is perhaps expected, as the PolyU dataset has lower noise density and a simpler distribution, which inherently limits the potential for significant improvement compared to the SIDD dataset, particularly in fully unseen settings.
>
> In the final version of our paper, we will ensure that your invaluable suggestions are fully incorporated. If you have any additional questions or concerns, we would be more than happy to address them. Please feel free to leave a comment at any time. Thank you for your thoughtful review and constructive feedback!
>
> Best regards, Authors.

---

### Official Review · Reviewer_s2ny · 2024-11-03

**Soundness:** 3
**Presentation:** 2
**Contribution:** 3
**Rating:** 6
**Confidence:** 3

**Summary:**

This paper presents an image restoration approach that utilizes diffusion models to address domain adaptation issues. The method adjusts restored results in the pixel-wise noise space, resulting in significant improvements in low-level visual appearance while operating within a compact and stable training framework. To avoid shortcut learning, the method employs channel index shuffling and a residual-swapping contrastive learning strategy. Experimental results demonstrate this approach outperforms existing feature-space and pixel-space domain adaptation methods across various tasks, such as image denoising, deraining, and deblurring.

**Strengths:**

1. The method enhances image restoration performance by effectively bridging the gap between synthetic and real-world data.

2. The approach can be easily integrated with various restoration networks, demonstrating improved performance even with more complex architectures.

**Weaknesses:**

1. The writing in this paper is not very clear, and I found the third section particularly confusing regarding how the proposed method leverages the mapping learned from synthetic datasets to be applied to real datasets. There must be some form of "information leakage" involved—this is a proactive "leakage" that helps bridge the gap between different domains. However, the authors do not provide a clear explanation of this process, and it appears that channel shuffling plays a key role in this function. A more explicit clarification from the authors would be beneficial.

2. The paper lacks runtime for the proposed method.

**Questions:**

Could the authors give concrete examples of situations where their proposed method fails to deliver the expected results?

I am impressed by the significant improvements reported by the authors. I wonder how the results would compare if the model were trained directly on a paired real dataset and then tested on corresponding real data.

I expect a direct and clear explanation from the authors regarding how the knowledge is "leaked" from the synthetic domain to the real domain in the proposed method.

---

> ### Author Response · Authors · 2024-11-22
> **Response to Reviewer s2ny (Part 1/2)**
>
> > ### W1. An explanation for "information leakage" involved in the training stage
>
> We sincerely thank the reviewer for pointing out this interesting and insightful perspective of proactive "leakage". We would like to answer and discuss it from the following two aspects:
>
> - Our motivation for this work is derived from an interesting observation: the prediction error of a conditional diffusion model relies on the quality of the conditions (as shown in Fig. 1 (a)). Therefore, guided by the back-propagated diffusion loss, the restoration network is optimized to provide "good" conditions to minimize the diffusion model’s noise prediction error, aiming for a clean image distribution. During this joint training, the synthetic GT serves as the denoising target in the diffusion model, which potentially offers realistic textures to help adapt the degraded real-world images into the clean distribution. In other words, the clean knowledge/information "leaked" by the diffusion’s input (in a multi-step denoising manner) plays an important role in bridging the gap between different domains.
>
> - Generally, the ground truth images and those restored images by a restoration model, whether from synthetic or real-world domains, should reside within a common distribution of high-quality, clean images. However, the appearance of synthetic GT and restored real-world data is unrelated, leading the diffusion model to exploit a shortcut, overfitting its denoising capability by relying solely on the paired synthetic data. To this end, we further design crucial strategies (*e.g.*, channel-shuffling layer and residual-swapping contrastive learning) to implicitly blur the boundaries between conditioned synthetic and real data and prevent the reliance of the model on easily distinguishable features. As a result, the proactive “leakage” from clean distribution to degraded images can effectively work during the whole training process, consistently improving the restoration performance on real-world images.
>
> > ### W2. Runtime for the proposed method
>
> As suggested, we have updated the runtime for comparison methods as follows. All methods are tested on an NVIDIA A100 40G GPU.
>
> | Task           | Resolution  | Method         | Time (s)    |
> |----------------|-------------|----------------|-------------|
> | Denoise        | 256x256     | Ours           | 0.0015    |
> |                |             | CyCADA         | 0.0015    |
> |                |             | Ne2Ne          | 0.0025    |
> |                |             | MaskedD        | 0.2418    |
> | Derain         | 512x512     | Ours           | 0.0020    |
> |                |             | NLCL           | 0.0139    |
> |                |             | Restormer      | 0.1561  |
> | Deblur         | 670x764     | Ours           | 0.0022    |
> |                |             | SelfDeblur     | 0.0147    |
> |                |             | VDIP           | 0.0102    |
>
> Compared with previous methods, our work can achieve promising restoration performance on real-world datasets while showing favorable efficiency. This advantage stems from the flexibility of the proposed domain adaptation strategy, which only requires the conditional diffusion model during the training stage. The diffusion model is discarded after training, incurring no extra computational cost in restoration inference.
>
> > ### Q1. Could the authors show some failure cases?
>
> Sure, the failure case has been added in the updated Appendix. Please refer to Fig. A9 for more details. In this case, we show our method failed to restore the images with challenging degraded distortions such as strong noises and out-of-distribution noises. These real-world degradations induce an extremely significant gap compared with the synthetic dataset, burdening the learning model to effectively adapt the restored results into the clean domain. Such a challenge also occurs on other comparison methods. It could be potentially addressed by incorporating more powerful diffusion models and we leave it as the future work.

---

> > ### Author Response · Authors · 2024-11-22
> > **Response to Reviewer s2ny (Part 2/2)**
> >
> > > ### Q2. The performance of the model is trained directly on a paired real dataset
> >
> > Following your suggestion, we have trained the denoiser model directly on a dataset with real paired samples; the evaluated performance (upperbound) on the SIDD test set is reported as follows. As we can observe, there is an observable domain gap between the synthetic dataset and real-world dataset. As a result, restoration models trained on synthetic images with previous conventional loss functions cannot escape from a dramatic drop in performance when applied to real-world domains. By contrast, our method can effectively bridge this gap by adapting the model in the noise space using a novel diffusion loss, which only requires the degradation observation of real-world images (without their paired clean labels).
> >
> > | Metric | Trained on Syn | Trained on Real | Ours |
> > |:--------:|:----------------:|:------------------------------:|:--------:|
> > | PSNR   | 26.58          | 39.18                       | 34.71  |
> > | SSIM   | 0.6132         | 0.9547                      | 0.9202 |
> >
> > > ### Q3.  A direct and clear explanation of "information leakage"
> >
> > As discussed in W1, we have explained how the knowledge is "leaked" from the synthetic domain to the real domain. Please refer to W1 for more details. We appreciate it again for your insightful perspective.

---

> > > ### Comment · Reviewer_s2ny · 2024-11-26
> > >
> > > The authors' rebuttal addresses most of my concerns and I raise my score to 6. However, as pointed out by Reviewer vDAP, the paper's structure makes it difficult to follow. I share this view and believe that the presentation of the contributions, method flow, and technical clarity still requires significant improvement and major revisions. While I have slightly raised my score, it is worth noting that the paper needs substantial revision to clearly and directly highlight its core contributions and innovations. If the AC or SPC ultimately decides to reject the paper, I have no objection.

---

> > > > ### Author Response · Authors · 2024-11-26
> > > > **Response to Reviewer s2ny**
> > > >
> > > > Thank you for raising the rating. We are happy most of your concerns have been addressed. Your invaluable suggestions will be carefully incorporated into the final version to further enhance the quality of our work.
> > > >
> > > > For your suggestion on technical clarity, we have also added the detailed workflows of the training and inference stages in Algorithm 1 and Algorithm 2 of the Appendix (page-27). Additionally, clear explanations of the "information leakage" have been presented in Section A5.2 (page-27). Please refer to the updated paper for more details.
> > > >
> > > > If the reviewer has any additional questions or concerns, we would be more than happy to address them. Please feel free to leave a comment here. Thank you.
> > > >
> > > > Best Regards, Authors

---

> > > > > ### Comment · Reviewer_s2ny · 2024-12-03
> > > > >
> > > > > Thanks for feedback. I keep the score as marginally above the acceptance threshold.

---

> > > > > > ### Author Response · Authors · 2024-12-03
> > > > > > **Response to Reviewer s2ny**
> > > > > >
> > > > > > Dear Reviewer s2ny,
> > > > > >
> > > > > > We sincerely appreciate your constructive comments and are encouraged by your positive score on our work. Your suggestions have contributed to significantly enhancing the quality of our paper.
> > > > > >
> > > > > > Best Regards, Authors

---

### Official Review · Reviewer_5su7 · 2024-11-04

**Soundness:** 3
**Presentation:** 3
**Contribution:** 3
**Rating:** 6
**Confidence:** 2

**Summary:**

The core of the method is a two-model setup: a restoration network and a diffusion model. The restoration network is trained to clean degraded images, while the diffusion model acts as a noise-space guide, enforcing an alignment between synthetic and real-world restorations and the clean target distribution. The authors validate their approach on tasks such as image denoising, deblurring, and deraining, showing it consistently outperforms both feature-space and pixel-space domain adaptation methods.

**Strengths:**

The strengths of this work lie in the following:
- A novel approach for iteratively using a diffusion model as a proxy model for denoising in noise space
- The authors introduce channel-shuffling and residual-swapping strategies, ensuring robust generalization.
- The diffusion model is used only during training
- The final restoration model achieves SOTA results across multiple restoration tasks.

**Weaknesses:**

For de-raining results, the authors should show results where the background is not empty, e.g. where there is more content in the image besides a gray background as in Figure 5.

**Questions:**

Can the method be trained with multiple tasks simultaneously since it is technically (low-level) task agnostic?

---

> ### Author Response · Authors · 2024-11-22
> **Response to Reviewer 5su7**
>
> > ### W1.  Show results where the background is not empty in de-raining images
>
> Thanks for your suggestion. We have updated the de-raining results where the background is not empty. Besides, we also provided more de-raining results under various scenes in Fig. 8, Fig. A4 and Fig. A8. Please refer to the updated paper for more details.
>
> > ### Q1. Can the method be trained with multiple tasks simultaneously since it is technically (low-level) task agnostic?
>
> We believe our work can be extended into such an application since the proposed framework offers a general and flexible adaptation strategy applicable beyond specific restoration tasks. It requires no prior knowledge of noise distribution or degradation models, and thus differs from self-supervised restoration methods tailored to one type of low-level task. We would like to leave this “all-in-one” network as our future work.

---

> ### Author Response · Authors · 2024-11-26
> **Please kindly let us know if your concerns have been addressed**
>
> Dear Reviewer 5su7,
>
> Thank you again for your review. We appreciate your recognition on our novelty and SOTA performance. We truly hope that our rebuttal can address your questions and concerns, such as showing results with more content and discussion on the "all-in-one" restoration model. As the discussion phase is nearing its end, we would be grateful to hear your feedback and wonder if you might still have any concerns we could address.
>
> It would be appreciated if you could raise your score on our paper if we address your concerns. We thank you again for your effort in reviewing our paper.
>
> Best Regards, Authors

---

> > ### Comment · Reviewer_5su7 · 2024-11-26
> >
> > Thank you for addressing my concerns. I agree with the other reviewers that there is clearly novelty and impact, but portions of the writing made it challenging to follow. I maintain my rating.

---

> > > ### Author Response · Authors · 2024-11-27
> > > **Response to Reviewer 5su7**
> > >
> > > Dear Reviewer 5su7,
> > >
> > > We sincerely appreciate your recognition of the clear novelty and impact of our work.  In the final version, we will ensure that all your suggestions are fully incorporated into the paper.
> > >
> > > For your suggestions on the paper writing, we have reorganized the structure and highlighted the proposed contributions in the paper.  Specifically, we have added the detailed workflows of the training and inference stages in Algorithm 1 and Algorithm 2 of the Appendix (page 27). Additionally, clearer explanations of bridging the domain gap have been presented in Section A5.2 (page 27). Please refer to the updated paper for more details.
> > >
> > > Thank you once again for your time and effort in reviewing our paper.
> > >
> > > Best Regards, Authors

---

### Meta-Review · Area_Chair_QuWH · 2024-12-19

**Metareview:**

This submission addresses image restoration tasks via domain adaptation in the noise space and proposes strategies to mitigate related shortcut-learning issues.

The authors performed exceptionally well in their rebuttal, addressing most concerns by presenting additional evaluation results and explanations. Although three reviewers initially leaned towards a negative view, they increased their scores after extended discussion with the authors.

After reviewing the paper, rebuttal and resulting discussions AC believes that the overall strengths outweigh the weaknesses and recommends acceptance. Employing domain adaptation in the noise space can be considered a valuable contribution to learning-based image restoration. For the camera-ready version, the authors should incorporate all key results presented in the rebuttal and significantly improve presentation aspects.

**Additional Comments On Reviewer Discussion:**

The paper received reviews from five reviewers: four borderline accepts and one borderline reject.

The authors performed exceptionally well in their rebuttal, addressing most concerns by presenting additional evaluation results and explanations. Although three reviewers initially leaned towards a negative view, they increased their scores after extended discussion with the authors.

Most reviewers recognise the significance of employing domain adaptation in the noise space, which can be considered a valuable contribution to learning-based image restoration and influenced the decision. The main concerns raised relate to: generalisation abilities with respect to unseen data, manuscript structure & writing quality and the comprehensiveness of experimental evaluation. The reviewer discussion helped to alleviate a subset of concerns raised.

---

### Decision · Program_Chairs · 2025-01-22

Accept (Poster)